# Parkinson's disease-linked Kir4.2 mutation R28C leads to loss of ion channel function

Xiaoyi Chen[1,2], Rocio K. Finol-Urdaneta[3] (iD), Mo Chen[1,4], Alex M. Sykes[1] (iD), Bingmiao Gao[5], Jamila Iqbal[1], David J. Adams[3] (iD), George D. Mellick[1,2] and Linlin Ma[1,2] (iD)

[1] *Institute for Biomedicine and Glycomics, Griffith University, Nathan, Queensland, Australia*
[2] *School of Environment and Science, Griffith University, Nathan, Queensland, Australia*
[3] *School of Medical, Indigenous and Health Sciences, Molecular Horizons, University of Wollongong, New South Wales, Australia*
[4] *Clem Jones Centre for Neurobiology and Stem Cell Research, Griffith University, Nathan, Queensland, Australia*
[5] *Engineering Research Center of Tropical Medicine Innovation and Transformation, Ministry of Education, Hainan Key Laboratory for Research and Development of Tropical Herbs, School of Pharmacy, Hainan Medical University, Haikou, China*

Handling Editors: Peying Fong & I-Shan Chen

The peer review history is available in the Supporting Information section of this article (https://doi.org/10.1113/JP287046#support-information-section).

The Journal of Physiology

**Abstract figure legend** Pathogenic impact of the PD-linked Kir4.2[R28C] mutation on Kir4.2 channel proteostasis and function. The Kir4.2[R28C] mutation, identified in a familial Parkinson's disease (PD) pedigree, leads to a near-complete loss of potassium channel function and exerts a significant dominant-negative effect. Wild-type Kir4.2 (Kir4.2[WT]) is synthesized in the endoplasmic reticulum (ER), undergoes glycosylation and proper folding and is trafficked to

This article was first published as a preprint. Chen X, Finol-Urdaneta RK, Chen M, Skye A, Gao B, Adams DJ, Mellick GD, Ma L. 2024. Parkinson.s Disease-Linked Kir4.2 Mutation R28C Leads to Loss of Ion Channel Function. bioRxiv. https://doi.org/10.1101/2024.05.05.592599

The Journal of Physiology

the plasma membrane, where it forms functional potassium channels that mediate $K^+$ conductance. A proportion of Kir4.2$^{WT}$ is also degraded via the lysosomal pathway. In contrast, the mutant Kir4.2$^{R28C}$ exhibits impaired protein stability and maturation, resulting in reduced overall protein levels. The mutant channels are inefficiently trafficked to the plasma membrane and are unable to form functional channels, thereby disrupting potassium homeostasis. This combination of loss-of-function and dominant-negative effects may contribute to the molecular pathogenesis of PD.

**Abstract** Parkinson's disease (PD) is a complex, progressive neurodegenerative disorder driven by multiple pathogenetic factors, including oxidative stress, mitochondria dysfunction, neuro-inflammation and ion imbalance. Recent evidence highlights the significant role of potassium channels in the pathophysiology of PD. We recently identified a PD-linked genetic mutation in the *KCNJ15* gene (*KCNJ15*$^{p.R28C}$), encoding the inwardly rectifying potassium channel Kir4.2, within a four-generation family with familial PD. However, the role of the Kir4.2 channel in neurodegenerative diseases remains largely unexplored. This study aimed to elucidate the impact of the *KCNJ15*$^{p.R28C}$ (Kir4.2$^{R28C}$) mutation on the biophysical and biochemical properties of Kir4.2. Employing Kir4.2-overexpressing HEK293T cells as a model, we investigated how the mutation affects the channel's functional properties, total protein expression, intracellular processing in the endoplasmic reticulum and lysosomes and plasma membrane trafficking. Patch clamp studies revealed that the Kir4.2$^{R28C}$ mutation results in loss of channel function with significant dominant-negative effects. This dysfunction is partially attributed to the substantial reduction in overall mutant channel protein expression compared to the wild-type (Kir4.2$^{WT}$). We observed that both Kir4.2$^{WT}$ and Kir4.2$^{R28C}$ proteins undergo glycosylation during the post-translational modification process, albeit with differing protein turnover efficiencies. Furthermore, the Kir4.2$^{R28C}$ mutant exhibits reduced stability and compromised plasma membrane trafficking capacity compared to Kir4.2$^{WT}$. These findings suggest that the Kir4.2$^{R28C}$ mutant has unique biomolecular and biophysical characteristics distinct from the Kir4.2$^{WT}$ channel, which potentially elucidates its role in the pathogenesis of PD.

(Received 4 June 2024; accepted after revision 22 April 2025; first published online 25 June 2025)

**Corresponding authors** D. J. Adams: School of Medical, Indigenous and Health Sciences, Molecular Horizons, University of Wollongong, NSW, Australia.    Email: djadams@uow.edu.au

G. D. Mellick and L. Ma: Institute for Biomedicine and Glycomics, Griffith University, Nathan, QLD, Australia.    Emails: g.mellick@griffith.edu.au and linlin.ma@griffith.edu.au

**Key points**

- Inwardly rectifying potassium channels are increasingly recognized for their critical role in the complex pathogenesis of Parkinson's disease (PD).
- We previously identified a genetic mutation, Kir4.2$^{R28C}$, in the inwardly rectifying potassium channel Kir4.2, which strongly segregates with familial PD in a multi-generational pedigree.
- This study confirms Kir4.2$^{R28C}$ as a loss-of-function mutation with significant dominant-negative effects, impairing channel activity even in heterozygous conditions.
- The Kir4.2$^{R28C}$ mutation significantly reduces overall protein levels, impairs protein stability and disrupts plasma membrane trafficking in *in vitro* cell models.

# Introduction

Parkinson's disease (PD) is a complex and multifactorial neurodegenerative disorder characterized by the progressive degeneration of dopaminergic neurons in the substantia nigra pars compacta (SNpc) region of the midbrain, often accompanied by the presence of Lewy bodies throughout the central nervous system (Bloem et al., 2021; Gibb & Lees, 1988). Several factors have been implicated as potential risk factors for PD including ageing, exposure to environmental toxins, oxidative stress, dysfunctions in mitochondrial and lysosomal processes and ion imbalance (Goldman 2014; Rodriguez et al., 2015). These factors can interact and create a cascade

of cellular dysfunctions, increasing the vulnerability of dopaminergic (DA) neurons to neurodegeneration. DA neurons are known for their intrinsic pacemaker activity, a self-generated electrical rhythm crucial for their function, dependent on the coordinated activities of various ion channels including $Na^+$, $Ca^{2+}$ and $K^+$ channels (Cantrell & Catterall, 2001; Kofuji & Newman, 2004; Pietrobon 2002; Vaidya et al., 2024). Particularly, the pathogenetic significance of $K^+$ channels and $K^+$ homeostasis in PD is gaining momentum (Chen et al., 2023).

Inwardly rectifying potassium (Kir) channels are essential for maintaining $K^+$ balance, regulating the duration of action potentials and influencing the resting membrane potential (Baronas & Kurata, 2014; Chen et al., 2023; Hibino et al., 2010). They are classified into four sub-families: G protein-coupled (GIRK or Kir 3.x), ATP-sensitive ($K_{ATP}$ or Kir6.x), classic (Kir2.x), and $K^+$ transport (Kir1.x; Kir4.x; Kir5.x and Kir7.x). Kir channels have distinctive structural and functional properties (Hibino et al., 2010). Several Kir channel family members have been demonstrated to play a critical role in PD pathogenesis (Chen et al., 2023). For example, the *Weaver* mouse model, characterized by the naturally occurring homozygous mutation p.G156S in the GIRK2 (Kir3.2) channel, exhibits a PD-like phenotype and has become an invaluable research tool extensively used in PD studies (Slesinger et al., 1996). This mutation converts GIRK2 from being highly selective for $K^+$ into a non-selective cation channel, resulting in increased excitotoxic stress in neurons, microglia-mediated neuroinflammation and subsequent neurodegeneration (Heginbotham et al., 1992; Patil et al., 1995; Peng et al., 2006; Schmidt et al., 1982).

In a recent genetic linkage study employing whole exosome sequencing, we discovered a novel genetic variant, *KCNJ15*[p.R28C], that segregates strongly with PD in a large family with ten diagnosed PD patients across four generations. This variant was detected in five PD cases and two unaffected individuals over the age of 80 within the family. Additionally, the same variant was found in other PD cases, including an isolated familial case in Italy with disease onset at 63 years and a sporadic PD patient in Australia with onset at 51 years (Bentley et al., 2021). This genetic evidence strongly suggests that the *KCNJ15*-encoded protein, the inwardly rectifying potassium channel Kir4.2, may play a significant role in PD pathobiology, similar to other well-known genetic mutations such as *LRRK2*[p.G2019S], which contributes to disease with intermediate penetrance (Hernandez et al., 2016; Trinh et al., 2014). Interestingly, the *KCNJ15* gene was also found to be significantly associated with Alzheimer's disease (AD) in a large-scale genome-wide association study of AD, where *KCNJ15* was highly expressed in the immune system and involved in immune-related events (Zhou et al., 2018). Therefore, the Kir4.2 channel may represent a potential new molecular target for therapeutic intervention and biomarker development in neurodegenerative diseases.

The tetrameric Kir4.2 channel belongs to the $K^+$ transport channel sub family, with each subunit consisting of 375 amino acids forming two transmembrane helices (Gosset et al., 1997; Shuck et al., 1997). The activation of the Kir4.2 channel is regulated by extracellular $K^+$ levels and intracellular pH, with a single-channel conductance of $\sim$25 pS and high open probability ($P_o \approx 0.7$–0.9) when the homotetrameric Kir4.2 is expressed in *Xenopus* oocytes (Edvinsson et al., 2011; Pessia et al., 2001). Although Kir4.2 is broadly expressed in various organs including the kidney, stomach, pancreas and brain, its physiological functions remain largely underexplored (He et al., 2011; Lourdel et al., 2002; Okamoto et al., 2012; Wang et al., 2022). Kir4.2 has been shown to curtail insulin secretion in pancreatic $\beta$ cells (Okamoto et al., 2010, 2012), regulate gastric acid secretion (He et al., 2011; Yuan et al., 2015), and may play an active role in epilepsy (Wang et al., 2022) and wound healing through regulating galvanotaxis (Nakajima & Zhao, 2016; Nakajima et al., 2015). However, little is known about the biochemical properties of this channel protein or how they are affected by disease-associated mutations.

In this study, we investigated how the genetic variant Kir4.2[R28C] affects the biophysical and biochemical properties of the channel. Kir4.2[R28C] was found to be a loss-of-function mutation with a dominant-negative effect and our biochemical analyses revealed a significant reduction in the overall expression of Kir4.2[R28C] compared to Kir4.2[WT] in transiently transfected HEK293T cells. To elucidate the mechanism underlying this differential expression, we examined

**Xiaoyi Chen** is a PhD candidate at the Institute for Biomedicine and Glycomics (IBG) at Griffith University, Brisbane, Australia. He earned his Bachelor's degree (with Honours) in Biomolecular Science from Griffith University. His PhD research, supervised by Professor George Mellick and Dr Linlin Ma, focuses on the role of ion channels in the development and progression of Parkinson's disease. He is passionate about advancing the understanding of the pathological mechanism of Parkinson's disease and identifying novel therapeutic targets for its treatment.

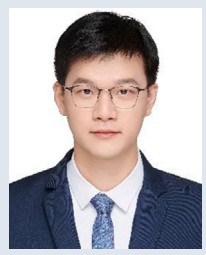

post-translational regulation, protein turnover in the cytoplasm, and the trafficking of the channel protein to the plasma membrane in detail.

## Methods

### DNA constructs and mutagenesis

The cDNA of $KCNJ15^{p.R28C}$ was cloned from the primary fibroblasts isolated from a mutation-carrying member in the family from which this new genetic variant was identified (Bentley et al., 2021). This cDNA was cloned into the pDONR201 vector (Thermo Fisher Scientific, Waltham, Massachusetts, USA) and subsequently sub-cloned into the pcDNA3.1-ccdB-3xFLAG-V5 vector (Addgene, Watertown, Massachusetts, USA) utilizing the Gateway cloning technique. To revert the $KCNJ15^{p.R28C}$ mutation to wild-type sequence, site-directed mutagenesis was performed using the pcDNA3.1-ccdB-3xFLAG-V5_ $KCNJ15^{p.R28C}$ construct DNA as a template. This process employed the KOD HiFi polymerase (Merck, Rahway, New Jersey, USA), following the provided instructions. Specifically, the reaction mixture consisted of 50 ng template DNA, 0.2 mM dNTPs, 0.4 µM of each forward (5′-CAACAGACCCTGCGTCATGTCCAAGAGTGGGC AC-3′) and reverse (5′-TTGGACATGACGCAGGGTCT GTTGGCCTTGAGCC-3′) primer, 1 mM $MgCl_2$, 1 U KOD HiFi PCR polymerase, and 1× KOD buffer. PCR conditions were set as follows: an initial denaturation at 95°C for 2 min, followed by 18 cycles of 95°C for 20 s, 55°C for 10 s, and 70°C for 130 s, with a final extension at 70°C for 5 min. Mutagenic PCR products were then digested with 10 U DpnI (New England Biolabs) at 37°C for 30 min to eliminate template DNAs. Successful mutagenesis was verified by full-length sequencing at the Griffith University DNA Sequencing Facility. Both $KCNJ15^{p.R28C}$ and $KCNJ15^{WT}$ cDNAs were then sub-cloned into pIRES2-eGFP vector for patch-clamp studies.

### Mammalian cell culture and transfection

HEK293T cells (ATCC, Manassas, Virginia, USA) were cultured in Dulbecco's-Modified Eagle's Medium (DMEM) (), supplemented with 10% fetal bovine serum (FBS) (Merck), 2 mM GlutaMax (Thermo Fisher Scientific), and 1× MEM NEAA (Life Technologies, Carlsbad, CA, USA) in a 37°C/5% $CO_2$ air-humidified incubator. For transfection, $10^5$ cells were plated in 500 µl of culture media per well in a 24-well plate. After overnight incubation, the cells were transfected with 500 ng plasmid DNA using Lipofectamine 2000 (Invitrogen) with a DNA:Lipofectamine ratio of 1:6 according to the manufacturer's protocol. With tunicamycin (Merck) or

bafilomycin A1 (Merck) treatment, transfected cells were exposed to the required concentrations for the specified duration before cell lysis.

### Patch-clamp electrophysiology

For patch-clamp studies, HEK293T cells were transiently transfected with Kir4.2$^{WT}$-pIRES2-eGFP and Kir4.2$^{R28C}$-pIRES2-eGFP constructs using Lipofectamine 2000 (Life Technologies) according to the manufacturer's instructions. At 18–24 h post-transfection, cells were mechanically dissociated, plated on glass coverslips and kept at 30°C with 5% $CO_2$ until recording.

Whole-cell currents were recorded from voltage-clamped cells using manual patch-clamp electrophysiology, following standard protocols (Bony et al., 2022; Finol-Urdaneta et al., 2022) using a Multiclamp 700B amplifier, with signals digitized via Digidata 1550 (Molecular Devices, San Jose, CA, USA) and analysed on a computer. GFP-positive transfected cells were identified under a Nikon Eclipse TS-100 fluorescence microscope before recording. Pipettes were pulled from GC 150 F-15 borosilicate glass capillaries (Harvard Apparatus, Kent, UK) in five stages and fire-polished to 2–3 MΩ resistance when filled with intracellular solution containing (in mM): 140 potassium gluconate, 10 NaCl, 2 $MgCl_2$, 5 EGTA and 10 HEPES, pH 7.2. Inward rectifier currents were recorded in a $K^+$-rich extracellular solution containing (in mM): 115 KCl, 30 NaCl, 1 $CaCl_2$, 1 $MgSO_4$, 5 glucose and 5 HEPES (pH 7.4).

Data were acquired and analysed using pClamp 10 software (Molecular Devices). Series resistance was maintained below 10 MΩ, and the cell capacitance was compensated by at least 70%. Membrane currents were sampled at 10 kHz and filtered at 1 kHz.

Inward rectifying $K^+$ currents were elicited using 400 ms voltage ramps from +150 to −150 mV (0.1 Hz) from a holding potential of 0 mV, recorded at room temperature (21–23°C) under control conditions and in the presence of 200 µM $Ba^{2+}$. Kir4.2 currents were determined as the difference between the $Ba^{2+}$-sensitive component, and the total inward current recorded at −150 mV. The resulting values were normalized to cell capacitance to obtain current density (pA/pF). Chord conductance (G) was estimated as:

$$G = \frac{I_{Ba^{2+}-sensitive}}{V - V_{\text{rev}}},$$

where $I_{Ba^{2+}-sensitive}$ represents the $Ba^{2+}$-sensitive currents between −20 and −140 mV, and $V_{\text{rev}}$ is the reversal potential.

## Western blot analysis

For immunoblot analysis, the cells in each well in a 24-well plate were harvested with 1000 µl ice-cold PBS and centrifuged at $300 \times g$ at 4°C for 10 min. The cell pellet was lysed using 200 µl of lysis buffer (50 mM Tris, pH = 7.4; 150 mM NaCl; 1 mM EDTA; 1× Complet Protease Inhibitor Cocktail (Roche, Basel, Switzerland) and PhosSTOP phosphatase inhibitor cocktail (Roche); 0.25% sodium deoxycholate; 0.1% SDS; 1% IGEPAL CA630). The cell lysate was homogenized with a Branson S450A Sonifier cell disruptor (Output 30%, duty cycle 20, 10 s for five times), followed by centrifugation at $14,000 \times g$ at 4°C for 10 min. The total protein concentrations were quantified using the Pierce BCA Protein Assay kit (Thermo Fisher Scientific) following the manufacturer's protocol. For SDS-PAGE, protein samples (20 µg) were mixed with 5× SDS sample loading buffer with 100 mM DTT, incubated at room temperature for 60 min, and stored at −20°C overnight. If required, protein samples were incubated with 500 U of PNGase F or Endo H (New England Biolabs, Ipswich, Massachusetts, USA) at room temperature for 4 h before being stored at −20°C overnight. The samples were centrifuged at $14,000 \times g$ for 30 s and loaded onto an SDS-PAGE gel (4% stacking gel and 12% resolving gel) in 1× SDS running buffer (25 mM Tris, 192 m glycine, 0.1% SDS). PageRuler Plus Pre-Stained Protein ladder (Thermo Fisher Scientific) was used to estimate molecular weight. The gel was run at 150 V for 1 h until the front of the dye had reached the bottom of the gel. Following SDS-PAGE separation, the proteins were transferred to a nitrocellulose membrane in a transfer buffer (25 mm Tris-base, 192 mm glycine, 15% (v/v) methanol) at 230 mA for 90 min. The membrane was blocked with a blocking buffer (5% BSA diluted in PBS-T (137 m NaCl, 2.7 m KCl, 10 m Na$_2$HPO$_4$, 1.8 m KH$_2$PO$_4$, 0.1% (w/v) Tween-20) or TBS-T (20 m Tris-base, 150 m NaCl, 0.1% (w/v) Tween-20) for 1 h. The membrane was then incubated with primary antibodies diluted in PBS-T or TBS-T at room temperature for 90 min and washed with PBS-T or TBS-T five times, for 5 min each time. The fluorophore-conjugated secondary antibodies were applied to the membrane for 1 h at room temperature. After the same washing procedure, the membranes were imaged using the Odyssey FC Imaging System (Li-COR, Lincoln, Nebraska, USA). The primary antibodies included rabbit mAb anti-V5-tag (D3H8Q) (Cell Signalling Technology, Cat # 13202S, 1:3000), rabbit mAb anti-Bip (C50B12) (Cell Signalling Technology, Danvers, Massachusetts, USA, Cat # 3177T, 1:3000), mouse mAb anti-$\alpha$-Tubulin (Merck, Cat # T5168, 1:24,000). The secondary antibodies included IRDye 680RD donkey anti-rabbit IgG (H + L) (LI-COR, Cat # 926–68073) and IRDye 800CW donkey anti-mouse IgG (H + L) (LI-COR, Cat # 925–32212).

## Immunofluorescence

For immunofluorescence experiments, transfected HEK293T cells were seeded on poly-L-ornithine (Merck, 0.0025%) pretreated coverslips at a density of 8000 cells/well (24 well) for 48 h. The cells were fixed with 4% paraformaldehyde (PFA) on ice for 10 min, blocked and permeabilized with 80 µl blocking solution (10% horse serum, 0.1% Triton X-100, diluted in PBS) for 1 h at room temperature. The primary antibodies (mouse mAb anti-FLAG M2 (Merck, Cat # F1804, 1:3000) and rabbit mAb anti-Na$^+$/K$^+$ ATPase (EP1845Y] (Abcam, Cat # ab76020, 1:3000) were applied to coverslips and incubated at 4°C overnight. The coverslips were washed with PBS before being incubated with secondary antibodies (Alexa Fluor 488 Donkey anti-Rabbit IgG (H+L) 1:1000, Alexa Fluor 647 donkey anti-mouse IgG (H+L), 1:1000) at room temperature for 90 min, and further treated with Hoechst (1:3000) at room temperature for 15 min before being mounted with ProLong Diamond Antifade Mountant (Thermo Fisher Scientific). The cells were imaged with the Olympus FV3000RS spectral confocal microscope using Olympus Fluoview software (Evident Scientific, Tokyo, Japan). Fluorescence intensities were quantified using Fiji (NIH, MD, USA) (Schindelin et al., 2012), CellProfiler (Stirling et al., 2021) and Rstudio (RStudio, PBC) (RStudioTeam 2020).

## Data analysis

Western blots were analyzed and quantified using ImageJ (Schindelin et al., 2012). The raw data were corrected for background subtraction before quantitative analysis. Signals were then normalized to the total protein loaded to each lane (20 µg/lane), with equal loading being determined using the housekeeping protein $\alpha$-tubulin.

Using the cell image analysis software CellProfiler (Broad Institute) (Stirling et al., 2021), several key steps were followed in analyzing immunofluorescence images. The process began with cell nucleus segmentation, where unnecessary image features from nuclear stains, such as nucleoli, were eliminated. After identifying the cell nuclei, which appear as complete ellipses, the cells' positions were determined using the nuclei's central coordinates, in a process known as cytoplasm segmentation. The cytoplasm size was then expanded from these central points based on the median cell size in the image. The third step involved cell membrane segmentation to accurately delineate the plasma membrane area. Despite using confocal microscopy for segmented scanning, signals from the cell membrane above and below the cell can show up as positive signals in the cytoplasmic region, complicating the accurate identification of the cell membrane position. After identifying the entire cell membrane's fluorescent signals, cytoplasmic regions were excluded to enhance

the depiction of the cell membrane area. Finally, to quantify the expression of Kir4.2 channels on the plasma membrane, the fluorescent signals from Kir4.2 within the clearly defined membrane area were normalized to the total membrane area and the total Kir4.2 signals from the cell, respectively.

Results for all experiments were generated from >3 independent biological repeats and presented as mean ± standard deviation. Student's unpaired *t* test or one-way ANOVA was used for statistical analysis. Data were considered statistically significant at $P < 0.05$ (*), $P < 0.01$ (**), $P < 0.001$ (***) and $P < 0.0001$ (****).

## Results

### *In silico* prediction of pathogenicity for the Kir4.2[R28C] mutation

Kir4.2 channel functions either as a homo- or a hetero-tetramer in conjunction with Kir5.1, with each subunit comprising two transmembrane helices. The ion-conducting pore is predicated to be formed by amino acids 126–132, as suggested by similarity analysis (Hibino et al., 2010; Pearson et al., 1999). As shown in the AlphaFold-predicted 3D structure of the human Kir4.2 channel (Fig. 1*A*), the amino acid change resulting from genetic variant p.R28C is located within the cytoplasmic N-terminus of the protein. Arg28 is poised to form hydrogen bonds with Asn37 and Glu304 (Fig. 1*B*). The mutation from the bulky, positively charged and less hydrophobic Arg (Fig. 1*C*) to smaller, more hydrophobic, neutral Cys (Fig. 1*D*) could disrupt these critical interactions, potentially affecting the protein's tertiary and quaternary structures.

To assess the potential impact of the Kir4.2[R28C] mutation, we conducted an *in silico* analysis using the bioinformatics platform HOPE (https://www3.cmbi. umcn.nl/hope/), which is a database for predicting the structural consequences of point mutations (Venselaar et al., 2010). The HOPE analysis assigned a MetaRNN score (ranging from 0.0 to 1.0) of 0.9605646 to Kir4.2[R28C], indicating that this mutation is highly likely to be pathogenic. This finding aligns with the assessment from another tool for scoring the deleteriousness of mutations, Combined Annotation Dependent Depletion (CADD, https://cadd.gs.washington.edu/), which assigned a Phred score of 16 to Kir4.2[R28C]. This score positions Kir4.2[R28C] among the top 2.5% of most deleterious substitutions, suggesting significant potential for pathogenicity (Rentzsch et al., 2019).

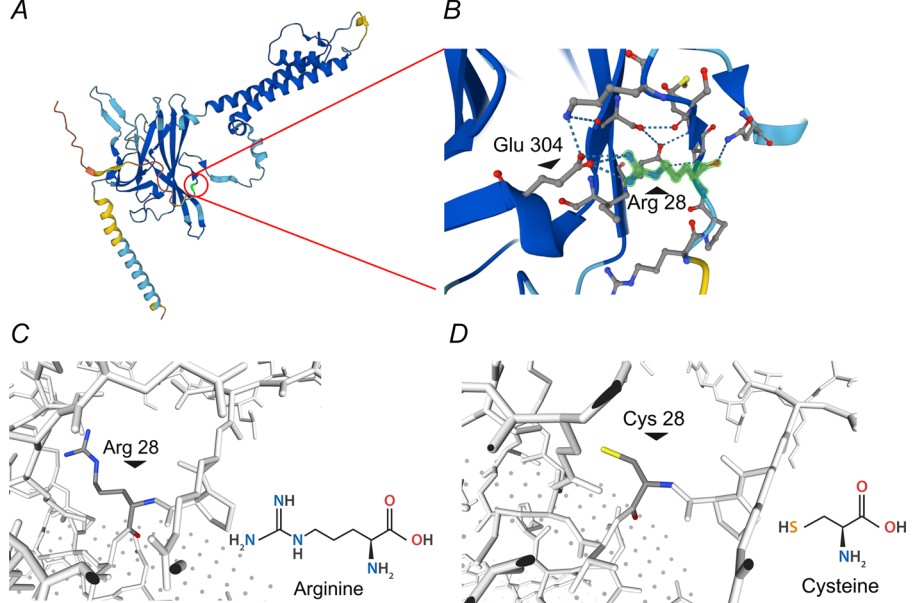

**Figure 1. Alphafold-predicted 3D structural model of the Kir4.2 channel**
*A*, predicted 3D structure of Kir4.2 channel protein (KCNJ15, accessible at https://alphafold.ebi.ac.uk/) (Varadi et al., 2022), illustrating a single subunit of the tetrameric channel. Arginine at position 28 (R28) is marked with a red circle. AlphaFold produces a per-residue confidence score (pLDDT) between 0 and 100. Blue pLDDT > 90; Cyan 90 > pLDDT > 70; Yellow 70 > pLDDT > 50; Red pLDDT < 50. The model's average confidence score is 0.85. *B*, a zoomed-in view of the highlighted R28 area and adjacent residues, with R28 emphasized in green. Dashed lines indicate potential hydrogen bonding. *C–D*, detailed visualization of the R28C mutation in Kir4.2 protein (accessible at SWISS-MODEL https://swissmodel.expasy.org/interactive) (Varadi et al., 2022; Waterhouse et al., 2018) with structures of the wild-type arginine (*C*) and the mutant residue cysteine (*D*) highlighted. In the highlighted amino acid, blue represents the amino group, yellow represents the thiol group and red represents the carbonyl group, respectively. [Colour figure can be viewed at wileyonlinelibrary.com]

## The Kir4.2$^{R28C}$ mutation results in loss of ion channel function with a dominant-negative effect

The functional significance of a genetic mutation in an ion channel protein is primarily determined by its impact on the ion-conducting properties of the channel. To assess the effect of the Kir4.2$^{R28C}$ mutation on channel function, we conducted whole-cell patch-clamp recordings in HEK293T cells. These cells were transiently transfected with either wild-type (Kir4.2$^{WT}$), the mutant Kir4.2$^{R28C}$ (MT), or a combination of both constructs (WT+MT). Inward rectifying K$^+$ currents were evoked by applying a 400 ms voltage ramp from +150 to −150 mV from a holding potential of 0 mV in a symmetrical K$^+$ solution (see Methods; Fig. 2).

Robust Ba$^{2+}$-sensitive inward currents were recorded at −150 mV in cells expressing Kir4.2$^{WT}$ channels (current

density: CD$^{WT}$ = −147.7 ± 23.1 pA/pF, N = 5), whereas cells expressing Kir4.2$^{R28C}$ channels showed a near-complete loss of Ba$^{2+}$-sensitive K$^+$ currents (CD$^{R28C}$ = −8.4 ± 2.3 pA/pF, N = 7) (Fig. 2A and B).

Strikingly, cells co-transfected with equimolar amounts of Kir4.2$^{WT}$ and Kir4.2$^{R28C}$ constructs exhibited significantly reduced Ba$^{2+}$-sensitive inward currents (CD$^{WT+MT}$ = −19.1 ± 3.2 pA/pF, N = 5) compared to those expressing homomeric Kir4.2$^{WT}$ alone (one-way ANOVA, P < 0.0001) (Fig. 2A and B). Chord conductance derived from ramp currents confirmed a significant decrease in total conductance in Kir4.2$^{WT+MT}$-expressing cells relative to Kir4.2$^{WT}$-transfected cells under standard recording conditions (Fig. 2C, P = 0.0006, two-way ANOVA). These findings demonstrate that the R28C mutation results in a loss-of-function phenotype, and

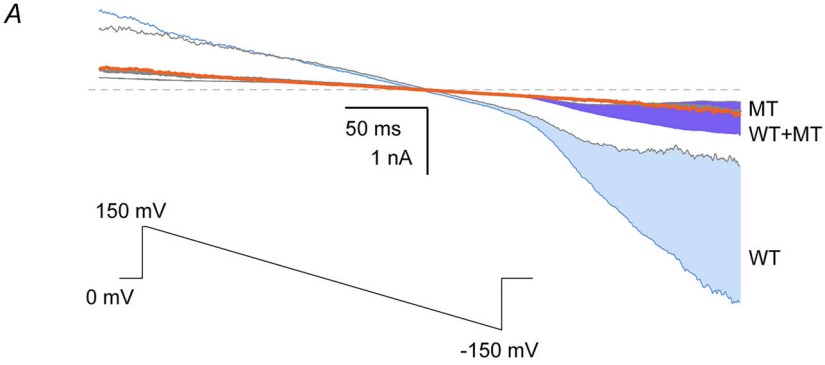

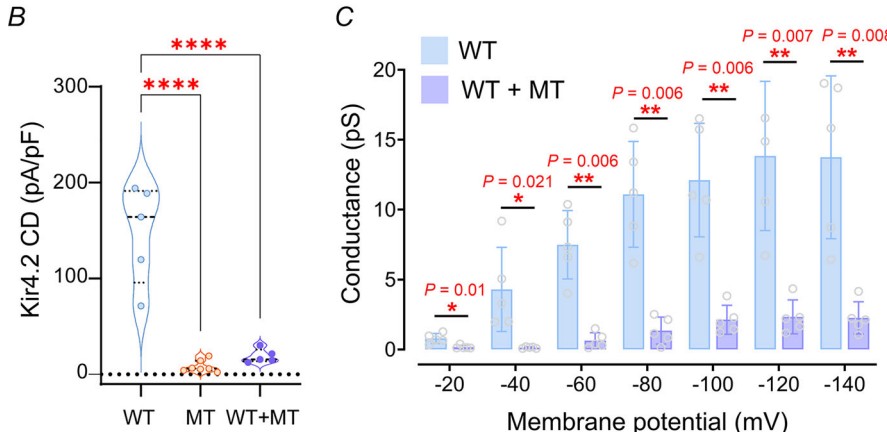

**Figure 2. The Kir4.2$^{R28C}$ mutation significantly reduces recombinant Kir4.2 channel mediated currents**
*A*, representative whole-cell current traces recorded from HEK293T cells transiently transfected with Kir4.2$^{WT}$, Kir4.2$^{R28C}$ (MT) and Kir4.2$^{WT+MT}$. The applied voltage protocol (inset) consisted of a 400 ms ramp from +150 to −150 mV, with a holding potential ($V_h$) of 0 mV. The Ba$^{2+}$-sensitive current components for cells over-expressing Kir4.2$^{WT}$, Kir4.2$^{R28C}$ (MT) and Kir4.2$^{WT+MT}$ are shaded in blue, orange and purple, respectively. *B*, Summary data are presented as a violin plot of current density (pA/pF measured at −150 mV) for Ba$^{2+}$-sensitive currents mediated by Kir4.2$^{WT}$ (N = 5), Kir4.2$^{R28C}$ (MT, N = 7), and co-expressed Kir4.2$^{WT+MT}$ (N = 5). Statistical analysis was conducted using one-way ANOVA (****P < 0.0001). *C*, chord conductance was derived from whole-cell ramp currents mediated by Kir4.2$^{WT}$ and Kir4.2$^{WT+WT}$ channels. Statistical analysis was conducted using multiple *t* tests. [Colour figure can be viewed at wileyonlinelibrary.com]

strongly suggest a dominant-negative effect of the Kir4.2$^{R28C}$ mutant on Kir4.2$^{WT}$ channel activity.

## The Kir4.2$^{R28C}$ mutation significantly reduces overall protein levels compared to Kir4.2$^{WT}$, with no impact on protein glycosylation

The Kir4.2$^{R28C}$ mutation, located on the cytoplasmic N-tail of the channel protein (Fig. 1), is unlikely to directly affect the ion selectivity or conductivity through the channel pore. To understand the mechanism underlying the loss of function caused by such a mutation, we first compared the overall expression levels of Kir4.2$^{WT}$ and Kir4.2$^{R28C}$ transiently transfected in HEK293T cells using western blot. Both Kir4.2$^{WT}$ and Kir4.2$^{R28C}$ exhibited two bands with apparent molecular weights of approximately 43 and 45 kDa, respectively. However, the overall protein level of Kir4.2$^{R28C}$ was consistently lower than that of Kir4.2$^{WT}$ across biological replicates (Fig. 3A). Quantitative analysis revealed that the overall expression level of Kir4.2$^{R28C}$ was less than 50% of that observed for Kir4.2$^{WT}$ ($P < 0.0001$, $N = 6$, Fig. 3B). This finding is noteworthy given that both Kir4.2$^{WT}$ and Kir4.2$^{R28C}$ are expressed under the control of the same promoter within the same expression vector. The observed disparity in overall protein levels is more likely attributable to differences in post-translational modulations and/or protein turnover rather than transcriptional regulation.

Kir4.2 channel has been predicted to undergo Asn-mediated N-linked glycosylation at positions 103 and 284 (Pearson et al., 1999; Shuck et al., 1997). Consequently, we proposed that the two distinct bands observed on the western blot represent different glycosylation states of the protein. N-linked glycosylation involves the attachment of a glycan to the nitrogen atom in the side chain of asparagine within the consensus sequence Asn-X-Ser/Thr (where X can be any amino acid except proline). This glycan is initially processed in the ER, where certain glucose and mannose residues are removed – a crucial step for the proper folding and quality control of the glycoprotein. Misfolded glycoproteins are targeted for degradation, whereas correctly folded ones are transported to the Golgi apparatus for further processing. Here, the glycan is extensively modified, including the addition and removal of sugar residues, which significantly impact the glycoprotein's final function, localization and stability (Fig. 3C).

To assess whether Kir4.2$^{R28C}$ exhibits the same glycosylation state and patterns as Kir4.2$^{WT}$, we treated the proteins with PNGase F and EndoH, respectively. PNGase F cleaves the bond between the innermost N-acetylglucosamine (GlcNAc) and the asparagine residue across all types of N-linked glycosylation. In contrast, Endo H targets the bond between the two

GlcNAc residues of high-mannose and some hybrid types of N-linked oligosaccharides but does not act on complex glycans. Therefore, glycoproteins processed in the early Golgi stages are sensitive to Endo H, whereas those processed in the later stages are resistant to Endo H (Fig. 3C) (Freeze & Kranz, 2010; Tarentino et al., 1985; Trimble & Tarentino, 1991). Treatment with PNGase F resulted in a single, lower molecular weight band for both Kir4.2$^{WT}$ and Kir4.2$^{R28C}$, representing fully deglycosylated proteins (Fig. 3D). Neither PNGase F nor Endo H altered the overall protein levels of Kir4.2$^{WT}$ or Kir4.2$^{R28C}$ (Fig. 3E). This indicates that removal of glycans from the protein does not impact the stability or solubility of Kir4.2, nor does it make the proteins susceptible to proteolysis that may happen during the experimental process.

In addition, a significant proportion of proteins shifted from the higher (upper band) to the lower (lower band) molecular weight band in both Kir4.2$^{WT}$ and Kir4.2$^{R28C}$, altering the upper/lower band ratios from >2 to <0.5 (Fig. 3D and F). This finding indicates that most Kir4.2 channels are not fully glycosylated within the cell. Furthermore, the similarity in upper/lower band ratios between Kir4.2$^{WT}$ and Kir4.2$^{R28C}$ suggests that Kir4.2$^{R28C}$ exhibits comparable glycosylation states and patterns to Kir4.2$^{WT}$.

## The stability and longevity of the Kir4.2$^{R28C}$ mutant are reduced relative to Kir4.2$^{WT}$

Both PNGase F and EndoH are widely utilized to treat purified proteins, making them useful tools for analysing the glycosylation state and patterns of protein products as static snapshots. However, they are not suitable for exploring the dynamics of protein glycosylation or its impact on protein stability over time within living cells. To investigate the effects of N-linked glycosylation on Kir4.2$^{WT}$ and Kir4.2$^{R28C}$ kinetics within cells, we treated cells overexpressing Kir4.2$^{WT}$ and Kir4.2$^{R28C}$ with tunicamycin before extracting and analysing the total proteins. Tunicamycin blocks N-linked glycosylation, potentially disrupting proper protein folding and function, thereby inducing cellular stress, the unfolded protein response (UPR) and the activation of stress pathways. Binding immunoglobulin protein (BiP), a marker of ER stress and UPR, served as a positive control (Fig. 4A), given its consistent and well-documented upregulation under such stress conditions across various cell types and experimental scenarios (Schroder & Kaufman, 2005).

Interestingly, Kir4.2$^{WT}$ and Kir4.2$^{R28C}$ exhibited differing responses to tunicamycin treatment (Fig. 4). Kir4.2$^{WT}$'s overall expression and the ratio between mature and immature proteins were not significantly

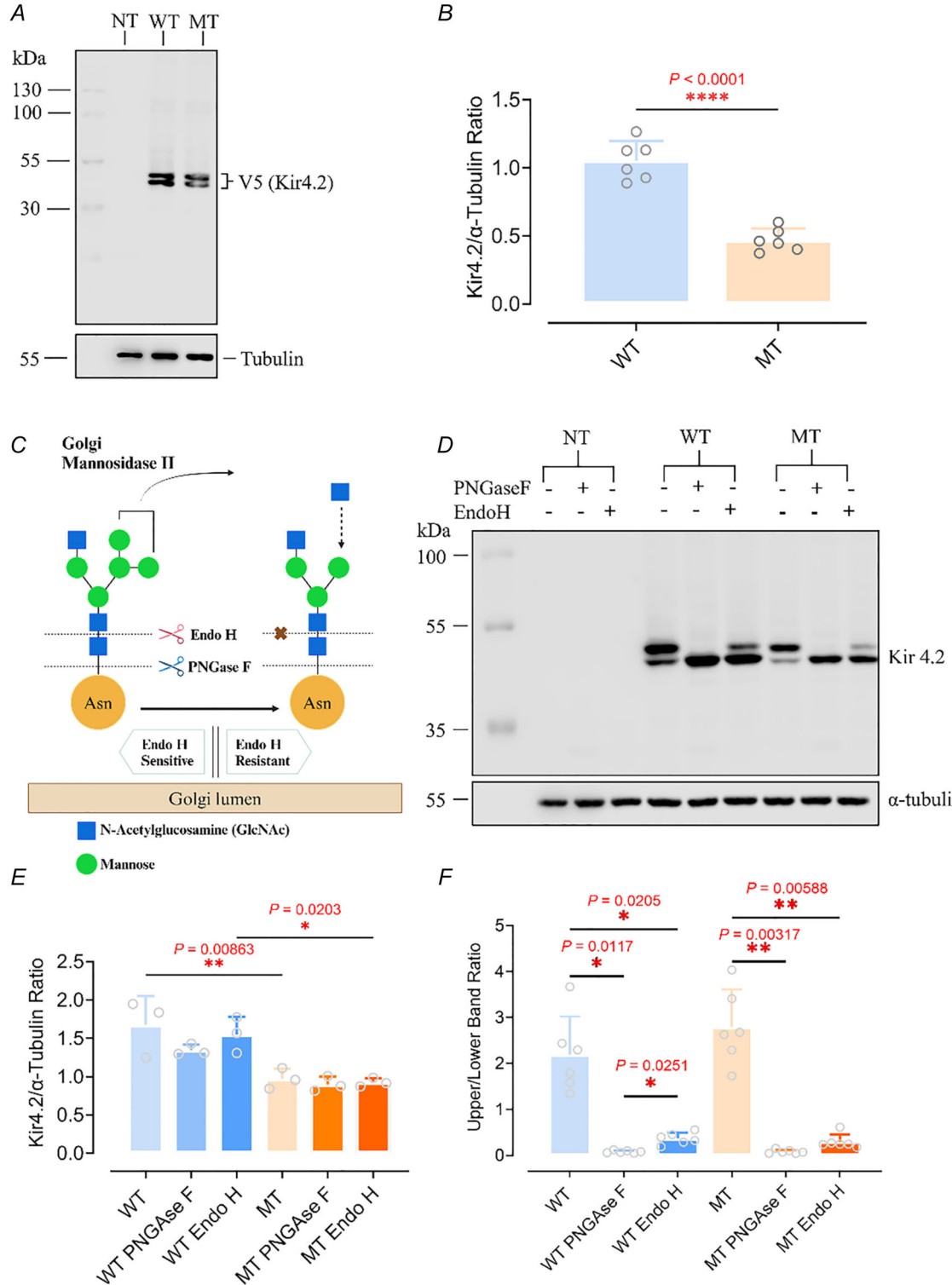

**Figure 3. Comparative analysis of Kir4.2$^{WT}$ and Kir4.2$^{R28C}$ in overall protein expression and glycosylation**

*A*, western blot analysis of protein expression levels of Kir4.2$^{WT}$ and Kir4.2$^{R28C}$, both of which were C-terminus fused with FLAG and V5 tags, in transiently transfected HEK293T cells using an anti-V5 antibody. NT, non-transfected; WT, Kir4.2$^{WT}$; MT, Kir4.2$^{R28C}$. *B*, quantitative analysis of Kir4.2$^{WT}$ and Kir4.2$^{R28C}$ protein levels using an unpaired *t* test (****$P < 0.0001$, $N = 6$). *C*, schematic illustration of the N-linked glycosylation process and the functions of enzymes PNGase F and Endo H. Created with BioRender.com. *D*, western blot for assessing the glycosylation states of Kir4.2$^{WT}$ and Kir4.2$^{R28C}$ through treatment with PNGase F/EndoH. *E* and *F*, quantitative

analysis of the overall glycosylation protein levels (*E*) and the ratio of mature (upper band) to immature (lower band) proteins (*F*) following treatment with PNGAse/Endo H, analysed by one-way ANOVA (**$P < 0.01$, *$P < 0.05$, $N = 3$–5). [Colour figure can be viewed at wileyonlinelibrary.com]

altered by tunicamycin (Fig. 4*B* and *C*). In contrast, Kir4.2$^{R28C}$ displayed a significant increase in overall protein levels following 6 h tunicamycin treatment ($P = 0.0446$, $n = 6$, Fig. 4*B*), along with a notable decrease in the proportion of mature (represented by the upper band) proteins ($P = 0.0031$, $N = 6$, Fig. 4*C*). The shift from mature (upper band) to immature (lower band) proteins was treatment time dependent ($P = 0.0129$ between 3 and 6 h of treatment, $N = 6$; Fig. 4*C*). Tunicamycin inhibits glycosylation by blocking the enzyme GlcNAc-1-P transferase, which catalyzMes the first step in the synthesis of N-linked glycan precursors; therefore it primarily affects new glycosylation events when applied to cells post-transfection (Heifetz et al., 1979; Wu et al., 2018). Given that both WT and mutant Kir4.2 proteins undergo N-glycosylation (Fig. 3*C* and *D*), these findings suggest that the WT proteins might be more resilient to disruptions in glycosylation, potentially due to inherent stability and/or a longer half-life compared to the mutant variants.

## The Kir4.2$^{R28C}$ mutant is less susceptible to autophagy-lysosomal degradation compared to Kir4.2$^{WT}$

Another crucial aspect influencing protein levels is protein degradation, an essential cellular function that manages protein turnover, ensures proteostasis, and eliminates damaged or misfolded proteins. Mammalian cells primarily utilize two pathways for protein degradation: the ubiquitin-proteasome system (UPS) and the autophagy-lysosomal pathway.

The UPS predominantly targets short-lived, misfolded or damaged proteins for degradation. It has been implicated in the regulation of the biogenesis of Kir channels, such as the $K_{ATP}$ channel (Steele et al., 2007; Yan et al., 2005). Within this system, proteins destined for degradation are tagged with polyubiquitins and subsequently recognized and dismantled by the 26S proteasome, liberating ubiquitin for future use and decomposing the target protein into peptides (Fig. 5*A*).

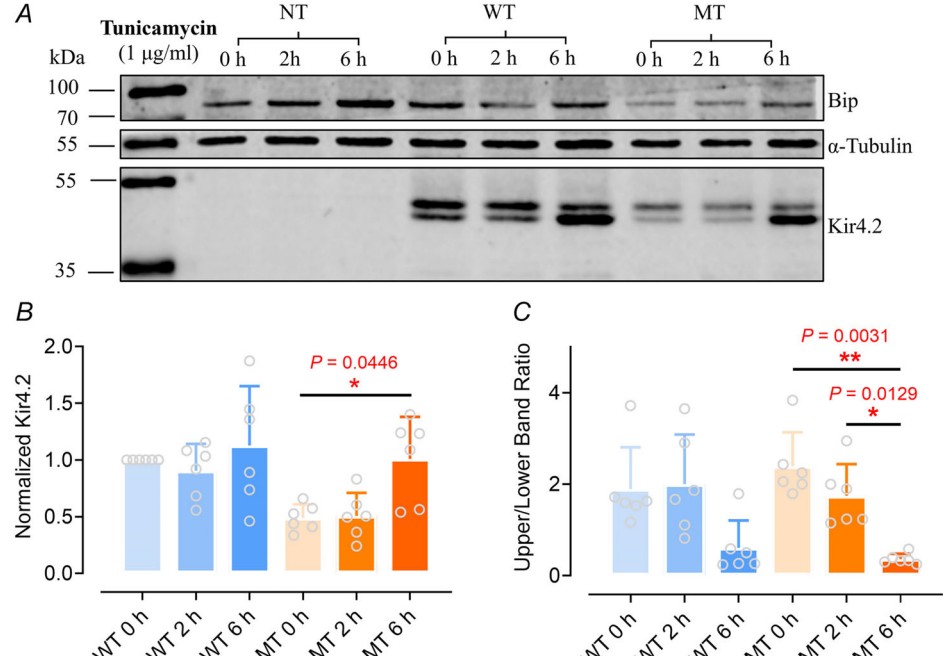

**Figure 4. Influence of endoplasmic reticulum stress on the Kir4.2$^{WT}$ and Kir4.2 $^{R28C}$ protein levels**
*A*, representative western blot analysis showcasing the effects of tunicamycin on Kir4.2 protein synthesis. Kir4.2$^{WT}$ and Kir4.2$^{R28C}$ transfected HEK293T cells were treated with 1 μg/ml tunicamycin for 0 h (negative control), 2 or 6 h prior to western blot analysis. NT, non-transfected; WT, Kir4.2$^{WT}$; MT, Kir4.2$^{R28C}$. Kir4.2 proteins were detected using an anti-V5 antibody. *B* and *C*, quantitative analysis of the overall Kir4.2$^{WT}$ and Kir4.2$^{R28C}$ protein levels normalized to the WT 0 h condition (*B*) and the ratio of mature (upper band) to immature (lower band) proteins (*C*) with or without tunicamycin treatment, using one-way ANOVA (**$P < 0.01$, *$P < 0.05$, $N = 6$). Binding immunoglobulin protein (BiP) served as a positive control for ER stress. [Colour figure can be viewed at wileyonlinelibrary.com]

To explore the potential regulation of Kir4.2 by the UPS, we employed the proteasome inhibitor MG132 to block UPS-mediated degradation (Fig. 5A) (Guo & Peng, 2013; Kim & Kim, 2020). MG132 was administered to HEK293T cells overexpressing Kir4.2[WT] and Kir4.2[R28C] 48 h post-transfection for 8 h (Fig. 5B). Despite 8 h MG132 treatment, the protein level of Kir4.2[R28C] remained significantly lower than that of Kir4.2[WT] ($P = 0.0483$, $N = 3$, Fig. 5B and C). Additionally, MG132 treatment did not significantly increase the protein levels of either Kir4.2[WT] or Kir4.2[R28C] (Fig. 5C), nor did it alter the ratio of mature to immature Kir channels (Fig. 5D). These results indicate that the UPS may not significantly influence Kir4.2 turnover within the cell.

The autophagy-lysosomal pathway is pivotal for degrading long-lived proteins, protein aggregates and damaged organelles. It encapsulates the target materials in double-membraned autophagosomes, which subsequently fuse with lysosomes to form autolysosomes that degrade the encapsulated materials (Fig. 5A) (Ballabio & Bonifacino, 2020; Dikic & Elazar, 2018). The lysosomal degradation of membrane proteins, including

Kir channels, has been well established (Estadella et al., 2020; Hager et al., 2021) (Fig. 5A; Jansen et al., 2008). Intriguingly, the Kir4.2 protein sequence harbours several LIR (LC3-Interacting Region) motifs, which bind to Atg8/LC3 protein family members to mediate autophagic processes, thereby tagging substrates for autophagy-lysosomal degradation (Geng & Klionsky, 2008; Kumar et al., 2022; Shpilka et al., 2011).

To investigate the role of lysosomal degradation in the differential expression levels between Kir4.2[WT] and Kir4.2[R28C], we administered bafilomycin A1 (Baf A1) to the cells. Baf A1 inhibits vacuolar-type $H^+$-ATPase (V-ATPase), effectively blocking lysosomal acidification and, consequently, autophagic flux, which is the process of autophagosome formation, fusion with lysosomes, and degradation of autophagic cargo (Klionsky et al., 2008; Yamamoto et al., 1998). As shown in Fig. 6, prolonged lysosome inhibition led to increased accumulation of Kir4.2[WT] proteins in the cells, suggesting that this channel protein is primarily degraded through the lysosomal or autophagic pathway ($P = 0.0211$ and $0.0042$ for 3 and 6 h treatment, respectively. $N = 8$, Fig. 6B). In contrast,

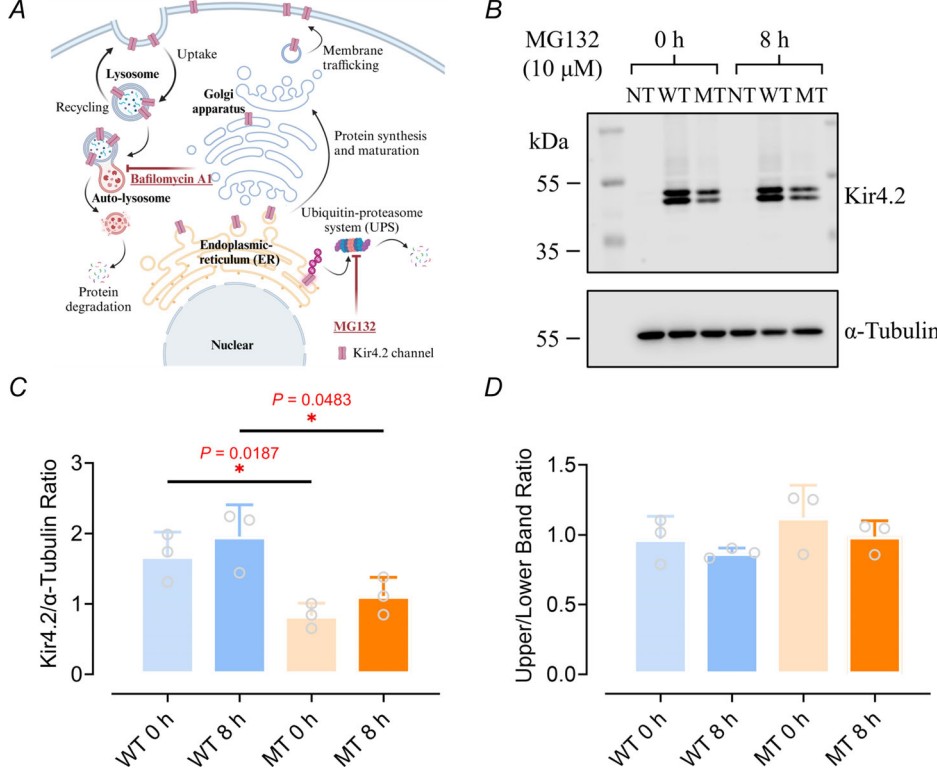

**Figure 5. Role of the ubiquitin-proteasome system in regulating Kir 4.2 channels**
*A*, schematic representation of cellular protein processing pathways and the mechanisms of action for MG132 and bafilomycin A1. Created with BioRender.com. *B*, western blot analysis of Kir4.2[WT] and Kir4.2[R28C] in HEK293T cells transiently transfected and treated with 10 μMM MG132 for 0 h (negative control) and 8 h. NT, non-transfected; WT, Kir4.2[WT]; MT, Kir4.2[R28C]. Kir4.2 proteins were detected using an anti-V5 antibody. *C* and *D*, quantitative evaluation of total Kir4.2[WT] and Kir4.2[R28C] protein levels (*C*) and the proportion of mature (upper band) to immature protein (lower band) forms (*D*) following MG132 treatment, assessed by unpaired *t* test (**P* < 0.05, *N* = 3). [Colour figure can be viewed at wileyonlinelibrary.com]

while Kir4.2$^{R28C}$ proteins exhibited a similar trend, they were less sensitive to Baf A1 treatment compared to Kir4.2$^{WT}$, indicating that this mutant may have additional regulatory controls that mitigate the effect of Baf A1 treatment (Fig. 6*B*). Furthermore, Baf A1 had similar effects on mature and immature Kir4.2 proteins, as the ratio of the upper to lower bands remained unchanged with the treatment for both Kir4.2$^{WT}$ and Kir4.2$^{R28C}$ (Fig. 6*C*).

### Kir4.2$^{R28C}$ exhibits impaired membrane trafficking

Kir4.2 channels perform their physiological functions at the plasma membrane, making the study of their membrane trafficking, especially in mutant forms, essential. We utilized immunofluorescence to analyze the trafficking of Kir4.2$^{WT}$ and Kir4.2$^{R28C}$ in HEK293T cells, as shown in Fig. 7*A* and *B*. The Na$^+$/K$^+$-ATPase, also known as the sodium-potassium pump, plays an essential role in maintaining homeostatic K$^+$ and Na$^+$ concentration gradients and is widely present in cell plasma membranes across various animal tissues. Due to its abundant expression at this location, it served as

a plasma membrane marker in this and many other biological and biochemical studies (Lobato-Álvarez et al., 2016; Schmid et al., 2022; Zhang et al., 2015). We observed that both Kir4.2$^{WT}$ and Kir4.2$^{R28C}$ channels are widely expressed within the cells, including at the plasma membrane. As expected, the overall expression levels of Kir4.2$^{R28C}$ were significantly lower than those of Kir4.2$^{WT}$ (Fig. 7*A* and *B*), aligning with our western blot findings (Fig. 3). Analysis of cells expressing either Kir4.2$^{WT}$ or Kir4.2$^{R28C}$ (Fig. 7*C*–*G*) revealed a significantly higher colocalization of Kir4.2$^{WT}$ with Na$^+$/K$^+$-ATPase at the plasma membrane than Kir4.2$^{R28C}$ ($P = 0.00984$, Fig. 7*H*). However, considering the notably higher overall expression of Kir4.2$^{WT}$ compared to Kir4.2$^{R28C}$ (Fig. 3), we sought to determine if the observed difference in membrane trafficking was solely attributable to differential expression levels. After normalizing the membrane trafficking data against the overall expression levels of Kir4.2, we found that the discrepancy between membrane-resident Kir4.2$^{WT}$ and Kir4.2$^{R28C}$ remained ($P = 0.0285$, Fig. 7*I*). These results require further verification using alternative methods. Nevertheless, current observations support the hypothesis that the mutation not only affects the stability and turnover of

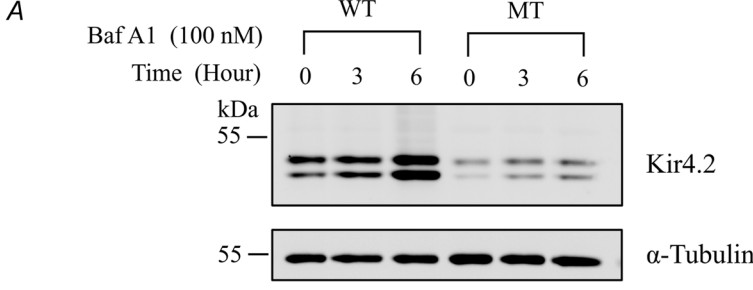

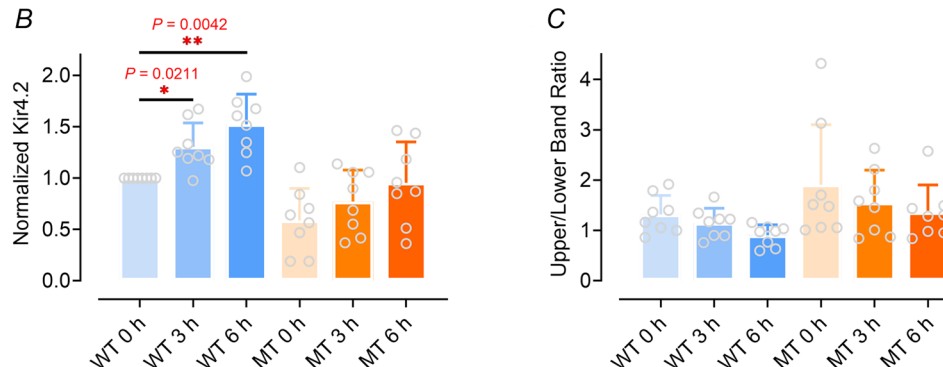

**Figure 6. Lysosomal degradation pathway regulates Kir4.2 channel processing**
*A*, western blot assessment of Kir4.2$^{WT}$ and Kir4.2$^{R28C}$ overexpressing HEK293T cells following treatment with 100 nM bafilomycin A1 (Baf A1) for 0 h (negative control), 3 h or 6 h. NT, non-transfected; WT, Kir4.2$^{WT}$; MT, Kir4.2$^{R28C}$. Kir4.2 proteins were detected using an anti-V5 antibody. *B*, total protein expression levels of Kir4.2$^{WT}$ and Kir4.2$^{R28C}$ were analyzed, with results normalized to the WT 0 h condition for comparison. *C*, analysis of the ratio between mature (upper band) and immature (lower band) protein forms following Baf A1 treatment, evaluated by one-way ANOVA (*N* = 8). LC3B served as a positive control for lysosomal activity. [Colour figure can be viewed at wileyonlinelibrary.com]

the Kir4.2 channel but also impairs the protein's plasma membrane trafficking capacity once it has been fully processed within the cell.

## Discussion

PD is a complex and multifaceted neurodegenerative disorder. While the precise origins of PD remain elusive, genetic factors have increasingly been recognized as playing a crucial role in its pathogenesis (Bloem et al., 2021; Gibb & Lees, 1988; Poewe et al., 2017). The identification of specific genetic mutations associated with both familial and sporadic cases of PD has significantly advanced our understanding of disease mechanisms (Bonifati et al., 1995; Lill 2016). Notably, mutations in genes such as *SNCA* ($\alpha$-synuclein) (Polymeropoulos et al., 1997), *PARK2* (Parkin) (Kitada et al., 1998), *PINK1* (PTEN-induced kinase 1) (Valente et al., 2001, 2004), *LRRK2* (Leucine-rich repeat kinase 2) (Funayama et al.,

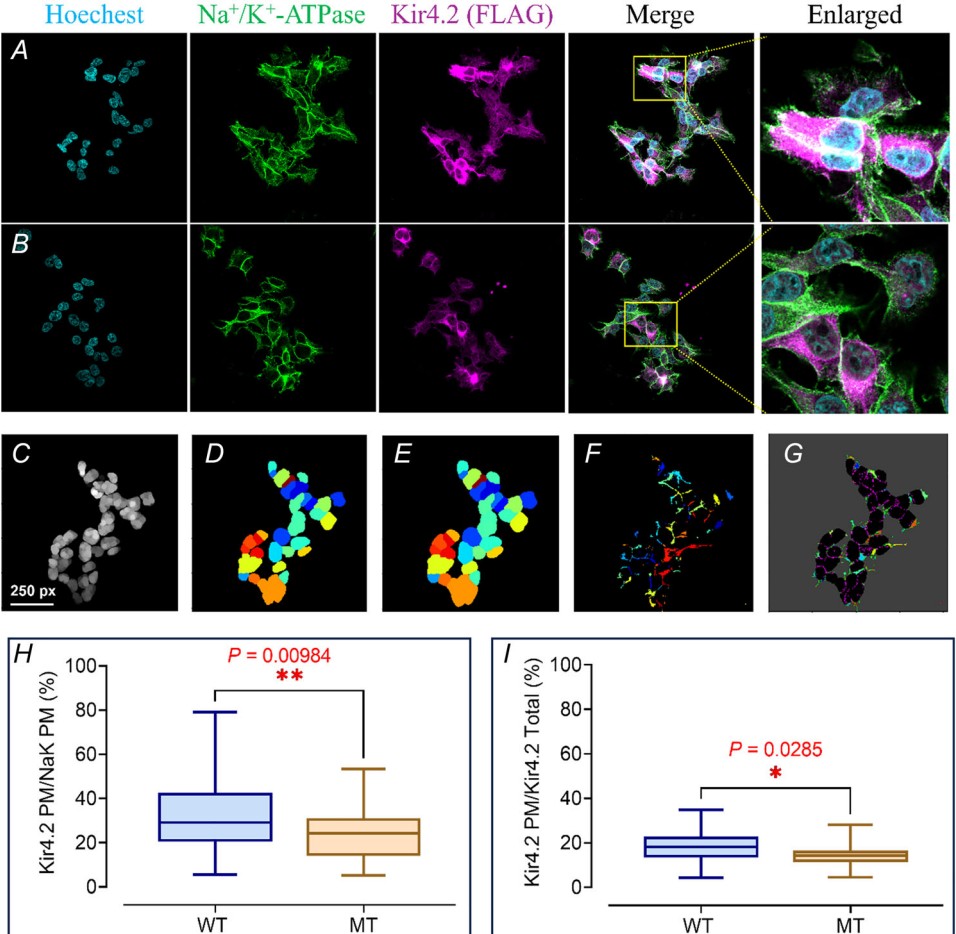

**Figure 7. The Kir4.2$^{R28C}$ mutation impairs plasma membrane trafficking capacity of the channel protein assessed by immunofluorescence analysis**

*A* and *B*, representative immunofluorescence analysis of HEK293T cells transiently transfected with Kir4.2$^{WT}$ (*A*) and Kir4.2$^{R28C}$ (*B*) using anti-FLAG antibodies for staining Kir4.2 proteins. Na$^+$/K$^+$-ATPase served as a marker for the plasma membrane. Images were acquired with an Olympus FV3000 confocal microscope at 60× magnification. *C–G*, quantitative analysis of Kir4.2 expression on the cell membrane involves (*C*) removal of redundant nuclear features through image processing techniques to enhance clarity, (*D*) recognition and segmentation of nuclear regions within the cell, (*E*) calculation of cytoplasmic area surrounding the nuclei, (*F*) precise identification of the cell membrane region and subsequent calculation of Kir4.2 expression area utilizing advanced image analysis algorithms and (*G*) refinement of analysis to specifically identify and quantify the Kir4.2 expression area localized on the cell membrane. *H* and *I*, comparative plot profile analysis of membrane trafficking between Kir4.2$^{WT}$ and Kir4.2$^{R28C}$. The membrane expression of Kir4.2 channels was normalized to the plasma membrane area defined by Na$^+$/K$^+$-ATPase signals (*H*) and total Kir4.2 expression throughout the cell (*I*), respectively. Statistical analysis was carried out using an unpaired *t* test (*$P < 0.05$, **$P < 0.01$). Data were from a total of 401 (Kir4.2$^{WT}$) and 399 (Kir4.2$^{R28C}$) positively transfected HEK293T cells, which were obtained from 45 (Kir4.2$^{WT}$) and 34 (Kir4.2$^{R28C}$) images captured from three biological replicates. [Colour figure can be viewed at wileyonlinelibrary.com]

2002), and *PARK7* (DJ-1) (Bonifati et al., 2003) have highlighted critical pathological mechanisms, including protein misfolding and aggregation, mitochondrial dysfunction, oxidative stress and neuroinflammatory pathways. These discoveries have not only facilitated the development of disease models but also paved the way for personalized medicine by enabling treatments tailored to individual genetic profiles (Blauwendraat et al., 2020; Lill 2016; Poewe et al., 2017).

Recent studies have increasingly implicated the involvement of potassium channels in PD pathogenesis (Chen et al., 2023). In addition to the well-characterized single mutation p.G156S in the GIRK2 (Kir3.2) channel, which induces parkinsonism in mice (Liss et al., 1999; Navarro et al., 1996), several other potassium channels including $K_V1.3$, $K_V2.1$ and Kir6.1, have been associated with PD (Chen et al., 2023). Notably, the recent identification of the Kir4.2$^{R28C}$ mutation in a PD-affected family (Bentley et al., 2021), highlights a potentially novel pathway involving $K^+$ homeostasis and neuronal signalling in PD pathogenesis.

Our electrophysiology analyses demonstrate the Kir4.2$^{R28C}$ homotetramers exhibit a complete loss of channel function (Fig. 2). When Kir4.2$^{WT}$ and Kir4.2$^{R28C}$ are co-expressed at an equal molar ratio, mimicking the heterozygous genotype observed in affected *KCNJ15* family members (Bentley et al., 2021), the resulting $K^+$ currents are markedly reduced (Fig. 2). This pronounced dominant-negative effect suggests that the mutant subunit interferes with the proper assembly or function of Kir4.2 tetramers, reinforcing its potential pathogenic role in both familial and sporadic PD cases.

*In vivo* studies using *KCNJ15*$^{-/-}$ mouse models suggest that Kir4.2 channels play a crucial role in motor and non-motor phenotypes (authors' unpublished data). Kir4.2 channels contribute to setting the resting membrane potential and regulate neuronal excitability. Loss of Kir4.2 function may lead to increased neuronal excitability, resulting in higher energy demands, elevated mitochondrial activity, and excessive production of reactive oxygen species, factors known to contribute to PD-related neurodegeneration (Drechsel & Patel, 2008; Orth & Schapira, 2002; Trist et al., 2019). Furthermore, increased neuronal excitability may trigger excessive glutamate release, leading to excitotoxicity, a process where overstimulation of neurons causes cell death (Caudle & Zhang, 2009; Vaarmann et al., 2013; Wang et al., 2020). Given that the basal ganglia, which regulate movement, rely on a balance between excitatory and inhibitory signals, diminished Kir4.2 function could disrupt this equilibrium, including chronic stress on neuronal networks and exacerbating PD's progression (Aron & Poldrack, 2006; Beste et al., 2010; Kumar et al., 2011).

Our *in vitro* analyses indicated that the Kir4.2$^{R28C}$ mutation exhibits significantly reduced expression compared to Kir4.2$^{WT}$ under identical experimental conditions (Fig. 3*A* and *B*). The expression of both Kir4.2$^{WT}$ and Kir4.2$^{R28C}$ was driven by the same promoter under identical conditions using model cell lines. Although a single mutation can potentially influence the transcription efficiency and translation efficiency under this condition, it is more likely that the observed differences in expression result from post-translational modifications, protein folding and turnover dynamics, which are critical for protein stability and function. Similar to the $\Delta$F508 mutation on the N-terminus of CFTR (Jensen et al., 1995; Lukacs et al., 1993, 1994), Kir4.2$^{R28C}$ also exhibits decreased stability and longevity compared to Kir4.2$^{WT}$ (Fig. 4). These findings suggest that the mutation might cause conformational instability, making it more prone to unfolding or partial unfolding under physiological conditions. The accumulation of misfolded or unstable proteins in the ER can trigger the UPR, a cellular stress response for restoring normal function by halting protein translation, increasing the production of molecular chaperones, and enhancing the degradation of misfolded proteins (Flick & Kaiser, 2012; Friedlander et al., 2000; Li et al., 2022). Additionally, misfolded proteins may aggregate and impair mitochondrial function (Conway et al., 1998; Cox et al., 2018; Tan et al., 2009), further exacerbating cellular stress and promoting degradation via the autophagy-lysosomal pathway (Flick & Kaiser, 2012; Li et al., 2009; Ohta et al., 2009). Our findings (Figs 5 and 6) support this, demonstrating that Kir4.2 proteins are predominantly cleared via autophagy-lysosomal system, rather than the ubiquitin-proteasome pathway.

Kir4.2 is a glycosylated protein (Fig. 3*D–F*), and glycosylation can enhance protein stability by preventing proteolytic degradation (Khanna et al., 2001; Lamothe et al., 2018; Russell et al., 2009). Additionally, glycosylation serves as a key signal for intracellular trafficking, ensuring correct localization of channels at the cell membrane (Baycin-Hizal et al., 2014; Fujita et al., 2006; Gong et al., 2002). Although the Kir4.2$^{R28C}$ mutant exhibits a glycosylation pattern similar to that of Kir4.2$^{WT}$ (Fig. 2*E* and *F*), the mutation significantly impairs plasma membrane trafficking (Fig. 7). Such trafficking defects are frequently observed in Kir channels. For instance, more than half of the missense mutations in Kir1.1 (ROMK) channel impair surface expression (O'Donnell et al., 2017; Peters et al., 2003). Similarly, analysis of mutation clusters within the C-terminal domains of Kir channels revealed that approximately 60% of mutations result in reduced membrane expression (Zangerl-Plessl et al., 2019). These defects may arise from defective ER export, altered Golgi processing, or misfolding-induced retention in intracellular compartments. Notably, the

residue corresponding to R28 in Kir4.2's homologue, Kir4.1 (R29), has been shown to contribute to an AP-1 clathrin adaptor-dependent Golgi export signal (27RRR29). Mutation of these three arginine residues on the N-terminus of Kir4.1 to alanine results in Golgi retention of the protein and significantly reduced membrane expression (Li et al., 2016). It would be of interest to investigate whether Kir4.2 harbours a similar clathrin adaptor-dependent Golgi export signal at its N-terminus as Kir4.1.

Nevertheless, the modest reduction in plasma membrane trafficking and the impaired stability of the mutant channel protein alone do not fully account for the complete loss of channel function. Although it may seem counterintuitive, mutations in the N-terminal cytoplasmic tail can profoundly affect channel function, as disruptions in gating and regulatory mechanisms are well documented across ion channel families. For instance, the ΔF508 mutation (deletion of Phe508) in the cystic fibrosis transmembrane conductance regulator (CFTR) protein disrupts folding and maturation, resulting in near-complete loss of function (Jensen et al., 1995; Lukacs et al., 1993, 1994). Similarly, the R104W mutation in the N-terminus of Nav1.5 abolishes $Na^+$ currents and exerts a dominant-negative effect, contributing to Brugada syndrome (Clatot et al., 2012; Doisne et al., 2021).

Notably, R28 is fully conserved across all Kir channels, underscoring its critical functional importance. A deep mutational scanning study of human Kir2.1 revealed that the equivalent R46C mutation markedly disrupts channel activity (Coyote-Maestas et al., 2022), consistent with structural evidence showing that R46 in Kir2.1 forms a stabilizing salt bridge between the N-terminal and the C-terminal cytoplasmic domains. Disrupting such inter-subunit interaction is known to impair channel function (Borschel et al., 2017). Further studies are required to elucidate the precise mechanisms by which the R28C mutation leads to loss of function in Kir4.2. In summary, this study investigates a novel Kir4.2 channel mutation identified in a large PD family spanning four generations. Our findings demonstrate that this mutation abolishes channel function, exerts dominant-negative effects, reduces overall expression and decreases protein stability. Additionally, the mutant channel exhibits impaired plasma membrane trafficking, which may exacerbate neuronal excitability and contribute to PD pathogenesis through mechanisms such as mitochondrial overactivity, elevated reactive oxygen species production and excitotoxicity. These insights not only enhance our understanding of PD but also suggest new therapeutic avenues, including strategies to stabilize Kir4.2 or regulate its degradation pathways.

Taken together, our findings highlight the Kir4.2[R28C] mutation as a potential contributor to PD through multiple mechanisms, including disrupted $K^+$ homeo-stasis, heightened neuronal excitability, mitochondrial dysfunction and impaired proteostasis. Further investigations are warranted to elucidate the precise role of Kir4.2 in PD and explore its viability as a therapeutic target.

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

## Additional information

### Data availability statement

The authors confirm that the data supporting the findings of this study are available within the article and available on request from the corresponding author.

### Competing interests

None declared.

## Author contributions

X.C. performed the molecular biology and cell biology experiments, analyzed data and drafted the original manuscript. R.K.F.-U. conducted the patch clamp studies and wrote the corresponding methods and results. M.C. designed and performed the immunofluorescence data analysis. A.S. and B.G. helped with the research design and resources. J.I. generated the pDONR201_KCNJ15p.R28C construct. D.J.A., G.D.M. and L.M. designed and supervised the research. All the authors contributed to the manuscript editing and have approved the final version of the manuscript. All persons designated as authors qualify for authorship, and all those who qualify for authorship are listed.

## Funding

This work was supported by the Michael J. Fox Foundation (MJFF-021285) to L.M. and G.D.M., and an NHMRC 2021 Genomics Health Futures Mission Stream 1 (2016760), and a NSW Cardiovascular Disease Senior Researcher Grant to D.J.A.

## Keywords

genetic mutation, inwardly rectifying potassium channel, Kir4.2, Parkinson's disease

## Supporting information

Additional supporting information can be found online in the Supporting Information section at the end of the HTML view of the article. Supporting information files available:

**Peer Review History**

