## [Peer Review History · The Journal of Physiology]

Parkinson's Disease-Linked Kir4.2 Mutation R28C Leads to Loss of Ion Channel Function

Xiaoyi Chen, Rocio K. Finol-Urdaneta, Mo Chen, Alex Sykes, Bingmiao Gao, Jamila Iqbal, David J. Adams, George D Mellick, and Linlin Ma

DOI: 10.1113/JP287046

Corresponding author(s): Linlin Ma (linlin.ma@griffith.edu.au)

Review Timeline:

Submission Date:	04-Jun-2024
Editorial Decision:	11-Jul-2024
Revision Received:	06-Mar-2025
Editorial Decision:	08-Apr-2025
Revision Received:	10-Apr-2025
Accepted:	22-Apr-2025

Senior Editor: Peking Fong

Reviewing Editor: I-Shan Chen

Transaction Report:

Dear Dr Ma,

Re: JP-RP-2024-287046 "Parkinson's Disease-Linked Kir4.2 Mutation R28C Leads to Loss of Ion Channel Function" by Xiaoyi Chen, Rocio K. Finol-Urdaneta, Mo Chen, Alex Sykes, Bingmiao Gao, Jamila Iqbal, David J. Adams, George D Mellick, and Linlin Ma

Thank you for submitting your manuscript to The Journal of Physiology. It has been assessed by a Reviewing Editor and by 3 expert referee and we are pleased to tell you that it is potentially acceptable for publication following satisfactory major revision.

LANGUAGE EDITING AND SUPPORT FOR PUBLICATION: If you would like help with English language editing, or other article preparation support, Wiley Editing Services offers expert help, including English Language Editing, as well as translation, manuscript formatting, and figure formatting at www.wileyauthors.com/eoo/preparation. You can also find resources for Preparing Your Article for general guidance about writing and preparing your manuscript at www.wileyauthors.com/eoo/prepresources.

REVISION CHECKLIST:

We look forward to receiving your revised submission.

Yours sincerely,

Peying Fong
Senior Editor
The Journal of Physiology

REQUIRED ITEMS

- Author photo and profile. First or joint first authors are asked to provide a short biography (no more than 100 words for one author or 150 words in total for joint first authors) and a portrait photograph. These should be uploaded and clearly labelled together in a Word document with the revised version of the manuscript. See Information for Authors for further details.
- Your manuscript must include a complete Additional Information section, including competing interests; funding; author contributions and acknowledgements.
- Please upload separate high-quality figure files via the submission form.
- You must upload original, uncropped western blot/gel images (including controls) if they are not included in the manuscript. This is to confirm that no inappropriate, unethical or misleading image manipulation has occurred. These should be uploaded as 'Supporting information for review process only'. Please label/highlight the original gels so that we can clearly see which sections/lanes have been used in the manuscript figures. For more information, see: <https://physoc.onlinelibrary.wiley.com/hub/journal-policies#imagmanip>.
- Please ensure that the Article File you upload is a Word file.
- Please include an Abstract Figure file, as well as the Figure Legend text within the main article file. The Abstract Figure is a piece of artwork designed to give readers an immediate understanding of the research and should summarise the main conclusions. If possible, the image should be easily 'readable' from left to right or top to bottom. It should show the physiological relevance of the manuscript so readers can assess the importance and content of its findings. Abstract Figures should not merely recapitulate other figures in the manuscript. Please try to keep the diagram as simple as possible and without superfluous information that may distract from the main conclusion(s). Abstract Figures must be provided by authors no later than the revised manuscript stage and should be uploaded as a separate file during online submission labelled as File Type 'Abstract Figure'. Please also ensure that you include the figure legend in the main article file. All Abstract Figures should be created using BioRender. Authors should use The Journal's premium BioRender account to export high-resolution images. Details on how to use and access the premium account are included as part of this email.
- Please include a full title page as part of your main article (Word) file, which should contain the following: title, authors, affiliations, corresponding author name and contact details, keywords, and running title.

Reviewing Editor's comments:

Your manuscript has been seen by three experts in the field. Referee #1 & #3 feel that this study provides novel information of the Kir4.2 mutation associated with PD. However, several points of critique are raised by referees. They suggest additional experiments to strengthen the conclusion (see Referee #1's comment on Figure 2 & 6 & 7, Referee 2's comment on Figure 7, and Referee #3's comment on Figure 4 & 6). Referee #2 request an explanation of how Kir4.2 mutant affects neurological functions or the degeneration of dopaminergic neurons if it doesn't express in either neurons or astrocytes.

Senior Editor's comments:

The initial review of your manuscript "Parkinson's Disease-Linked Kir4.2 Mutation R28C Leads to Loss of Ion Channel Function" now is complete.

Three subject experts concur that this study falls well within the remit of The Journal of Physiology, and that its conclusions hold promise for understanding how mutant R28C Kir4.2 channels participate in the etiology of Parkinson's disease. On the basis of its potential impact, I hope you take the opportunity to address the attached Referee's comments and resubmit a revised manuscript. As you read through these, please note that some require additional experiments to address convincingly, and on this basis can be considered major revision. To aid you in prioritizing your approach as you prepare your revision, please refer to the Reviewing Editor's succinct summary.

I have an additional request. I was unable to find information about a critical reagent used in the western blot analysis, i.e. the anti-Kir 4.2 (KCNJ15) antibody used in performing these blots. Information for such an antibody does show up in the Methods section pertaining to immunofluorescence studies. If the same antibody was used, please state this is the case within the preceding section on the immunoblotting/western blotting. This is important to know for robust data interpretation, since not all antibodies are useful for the detection across assay types. If you used the same antibody for both immunoblotting and immunofluorescence studies, please provide sufficient confirmatory data at the time you submit your revised manuscript as a supporting information file for review purposes only.

Referee #1:

In this manuscript, Chen et al., initially characterized an inwardly rectifying K⁺ channel decoded from KCNJ15p.R28C, a genetic mutation identified in patients with familial Parkinson's disease (PD), by patch-clamp approach. The authors found that this Kir4.2R28C channel, when expressed in HEK cells, elicited negligible current and showed a dominant-negative effect. Together with the data of biochemical and immunocytochemical assays, they concluded that loss of the channel's current is likely to result from protein instability associated with high susceptibility to lysosomal degradation pathway and reduction of plasma membrane trafficking capacity.

This work has approached the mechanisms underlying loss of function of the channel mutation associated with PD and provides beneficial information for not only experts who are involved in pathological processes of the disease but also non-experts who are interested in ion channels. To improve the manuscript, I have a couple of comments described below.

Major

1. From the data of double-immunolabeling described in Figure 7, the authors concluded that expression of Kir4.2R28C protein on the plasma membrane was reduced as compared to wild-type Kir4.2 and this change is a major source for loss of K⁺ current in the mutant channel. However, from my standpoint, the reduction of the surface protein is too modest (~10 %) to account for the dramatic suppression of the channel's current. Accordingly, the authors need to add multiple experiments to reinforce their conclusion. Minimum requirement is to isolate the membrane fraction and carry out western blotting with this sample.
2. In Figure 5, I can understand that MG132, a blocker for UPS-mediated degradation, had little effect on the protein level of Kir4.2WT and Kir4.2R28C. Nevertheless, before making such conclusion, the authors should examine if Kir4.2WT and Kir4.2R28C are ubiquitinated by biochemical approach with a suitable positive control such as KATP channel.
3. In Figure 6B and 6C, I am not convinced by the authors' conclusion described in the text because of large variation of the data for Kir4.2R28C treated with bafilomycinA1 for 6 h. To strengthen the conclusion, I request the authors to increase the data points (n = 6 at minimum).
4. As I mentioned in comment #1, there is a discrepancy between a dramatic loss of the current elicited by Kir4.2R28C channel (Figure 2) and a modest decrease of expression of the protein on the cell surface (Figure 7). This inconsistency suggests that the rest of the channel proteins on the cell surface would be characterized by impairment of single-channel properties. Therefore, the authors need to examine cell-attached patch to examine the change of the parameters such as open time and single-channel conductance.

Minor

1. In western blot analyses, two fragments are detected in both Kir4.2 WT and Kir4.2R28C. The terminology to indicate these two are confusing (upper/bottom or mature/immature). Throughout the manuscript, the authors should unify this issue.

Referee #2:

The paper deals with the effect of a mutation on the KCNJ15 potassium channel (R28C). The authors previously found the mutation associated with Parkinson's Disease in a few families. In this manuscript, the authors investigate the consequences of this mutation on the trafficking and stability of the channel proteins and its effects on function. KCNJ15-R28C is not functional as a homomer and is a dominant negative over the wt channel. This means the protein expresses and folds properly to allow assembly with the wt channel. Through elaborate biochemical assays, the authors conclude that the mutant displays reduced stability and is more prone to lysosomal degradation with reduced plasma membrane localization.

Major comments

The authors did not explain the dichotomy of how this channel mutant affects neurological functions or the degeneration of dopaminergic neurons if it doesn't express in either neurons or astrocytes. The brain atlas (<https://www.proteinatlas.org/ENSG00000157551-KCNJ15>) and other single-cell studies (<https://brainrnaseq.org/?532754659=2988537778>) do not provide evidence for the meaningful expression of the KCNJ15 channel in neuronal tissue or astrocytes, compared to channels that do express in neurons, like KCNJ6, or astrocytes, like KCNJ10.

In mice, this channel expresses mainly in kidneys proximal tubules to affect bicarbonate reabsorption (<https://doi.org/10.1016/j.kint.2019.09.028>) and glutaminase activity (<https://doi.org/10.1016/j.celrep.2022.111840>).

The authors should discuss the role of this residue in channel folding. There are ample structures of IR channels that can provide a good insight into the function of this fully conserved residue. Furthermore, R28 was identified as part of an AP-1 clathrin adaptor-dependent Golgi export signal (<https://doi.org/10.1074/jbc.M116.729822>); the authors should discuss their findings considering these findings.

Minor comments

It is very difficult to appreciate the expression pattern/level localized to the plasma membrane. Can the authors provide an expanded view of one cell in addition to the images in Fig. 7?

Referee #3:

Please see the attachment.

END OF COMMENTS

Overview

Chen et al. followed up their previous finding from extensive genetic analysis of large multi-incident families (Bantley 2021 Genes) that Kir4.2R28C mutation seems to be linked to Parkinson's disease (PD). This is a new finding, and it is important to elucidate how this mutation affects the channel activity in cells in order to understand the PD pathology. They used the heterologous expression system to characterize the mutation on channel function and found that the mutation showed a dominant negative effect on the co-transfected wild type proteins, and almost complete loss of function phenotype was observed for the homomeric mutant channels. The loss of function was partly explained by the reduced expression level of the mutant protein than the WT protein. They further interrogated intracellular trafficking and degradation of the mutant proteins. The results presented in the later parts of the manuscript clearly showed that the mutant proteins were expressed in the plasma membranes, and hence neither the reduced expression level nor the altered trafficking cannot solely explain the complete loss of function, which indicates that the mutant channel is likely functionally dead. However, this point has not been discussed. Also, it was claimed that Kir4.2R28C mutant behaves differently to drugs from the WT protein, and many arguments were made in the discussion, but the results are not very supportive of them. Clearly more repeats are required to get robust results, and the discussion should be toned down.

Major revision

Over- or miss-interpretation of the results is frequent throughout the manuscript;

Page 14 Line 1; "There was not statistical difference between the PNGase F-treated Kir4.2WT and Kir4.2R28C proteins, while with Endo H treatment, Kir4.2R28C exhibited a level approximately 50% of that observed for Kir4.2WT (Fig 3E)"

The experiment was done with harvested cells that were cultured under the same condition. Hence, if the protease inhibitor treatment was tight during the experiment, theoretically the total protein level of WT (and of R28C) among the three tested conditions should be identical. If they are different, it is due to protein degradation happened in the middle of the experiments or the analyses were not accurate. The difference mentioned in the above sentence arose due to the fact that the amount of WT protein in the PNGase F treated sample got decreased than the two other conditions (Non-treated and Endo H treated). Hence, the sentence does not support any mechanisms and is likely incorrect. So, it is recommended to exclude the sentence from the manuscript.

The important observations from this experiment (Fig. 3C-3F) is if the top bands shifted to the bottom ones after the treatment of each de-glycosylation enzyme. As the authors stated, there is no difference between the WT and the mutant.

Page 14. Last paragraph: “Interestingly, Kir4.2WT and Kir4.2 R28C exhibited differing responses to tunicamycin treatment. Kir4.2WT’s overall expression and the ratio between mature and immature proteins were not significantly altered by tunicamycin (Fig 4B&4C).”

The argument is not supported by the actual experimental data shown in Fig 4A. Fig 4A shows that 6h treatment tunicamycin caused the total expression level of both the WT and the mutant increased and the relative ratio between the top and the bottom band flipped, indicating newly synthesized proteins could not be glycosylated, which was the same for both proteins. (Also please see the minor revision point below regarding this figure.) Hence, the actual data is in contrast to their argument.

It seems like that one out of 3 replicate experiments showed no increase in the protein level and no increase in the un-glycosylated (immature) protein for the WT protein after 6hrs tunicamycin treatment. The one data point actually abolished the statistical significance. In my humble opinion, overall, the data indicate that both proteins responded to the tunicamycin treatment in the same manner. A few more replicates are needed to consolidate their argument, or the data figure should be replaced with the one matching the argument.

Page 17. Line 8 from the bottom: “Although not as pronounced as the increase in LC3B-II, a discernible trend showed that the overall levels of both Kir4.2WT and Kir4.2R28C rose with extended Baf A1 exposure (Fig. 6B). Notably, the baseline level of Kir4.2R28C was approximately 50% that of Kir4.2WT, yet this difference vanished following a 6-hour Baf A1 Treatment (Fig. 6B).”

This argument is not supported by the actual data shown in Fig 6A. essentially there is no changes in protein levels after Baf A1 treatment for both WT and R28C mutant proteins. More repeats are needed to support their argument, or the data figure should be replaced with the one matching the argument.

Figure 7. They used colocalization imaging to assess membrane expression of the proteins. But it may be more reliable and quantitative to use the biotinylation and membrane fractionation method.

Page 23, First paragraph: “Given that the expression of both Kir4.2WT and Kir4.2R28C was driven by the same promoter under identical conditions using model cell lines, it is more likely that this differential expression is caused by post-translational modifications, protein folding and turnover issues, which are crucial for the protein stability and function. ”.

This presumption is not automatically true since the protein level can be lower due to slowed translation and also reduced mRNA stability due to mutations. So, there are other mechanisms how protein level is kept lower than the other.

“Interestingly, similar to the dF508 mutation on the N-terminal tail of CFTR, the mutant channel Kir4.2R28C also exhibits decreased stability and longevity in cells compared to Kir4.2WT (Fig 4).”

I cannot figure how Fig 4 supports this argument. Could you elaborate?

Page 23. Second paragraph: “Indeed, we found that Kir4.2 proteins are not primarily subjected to the ubiquitine-proteasome system mediated degradation(Fig5), but are more susceptible to the autophagy-lysosomal degradation pathway (Fig 6).”

The first half is agreeable, but the second half is not. I cannot see any differences between the two proteins in Fig 6 in their response to the Baf A1 treatment.

Minor revision

Figure 4. It is unclear if the total expression time among the three conditions were identical for this experiment. Was the drug treatment started 6hrs, 2 hrs, and 0 hrs before the harvest so that the total length of time of protein expression is identical?

Since the experiment was done with transient transfection, the total amount of protein expression should increase as the incubation time elongates unless the treatment was done after the protein expression reached the steady state after the transfection. So I just want to double check if the increased total protein level of WT and R28C after 6 hour treatment was simply due to the cells were cultured for a longer time.

The tick label in Fig 4B and 4C should be fixed to be 2h instead of 3h.

Page 6: method for western blot: it should be 24-well plate instead of 25-well plate

Page 24: the last part of the first paragraph: “Particularly, the Kir4.2R28C mutant disrupts a PKB phosphorylation recognition site, a modification previously reported to affect the assembly and stability of the potassium channel/scaffold protein complex through phosphorylation.”

Tanemoto et al 2002 reports PKA phosphorylation of the serine residue at the very C-terminus of Kir5.1 affect its interaction with PSD95 proteins. Hence, it is hard to conceive how R28C

would disrupt PKB phosphorylation recognition site. Please remove this sentence or elaborate in detail how this argument arose.

Page 24: the last paragraph: “Our findings reveal that this mutation leads to abolished channel function, diminished overall expression, and decreased stability of the Kir4.2 protein. Furthermore, there is an increased likelihood of lysosomal degradation and compromised plasma membrane trafficking of the mutant channel.”

“This mutation leads to abolished channel function, diminished overall expression” this part is fully supported by the experimental results in the manuscript. However, the remaining part is not convincing. In order to get support for these arguments, more repeats are required for Fig 4 and 6. And Fig 7 can be further supported by the biotinylation and membrane fractionation experiment.

Professor Peying Fong
Senior Editor
Journal of Physiology

Re: Manuscript Number: JP-RP-2024-287046 "Parkinson's Disease-Linked Kir4.2 Mutation R28C Leads to Loss of Ion Channel Function"

Dear Professor Fong,

Thank you for your email regarding our submitted manuscript to the *Journal of Physiology*, and for recommending resubmission following major revisions. We apologize for the delay in resubmitting the revised manuscript, as the first author responsible for the required additional data was on leave for several months.

We sincerely appreciate the reviewers' thoughtful and constructive comments, which have significantly strengthened our manuscript. We are grateful for the time and effort invested in reviewing our work.

Our manuscript has now been extensively revised, addressing all the concerns raised. We fully align with the reviewers' intent and have carefully incorporated their feedback.

Please find below our detailed point-by-point response to their comments. We hope our manuscript will now be suitable for publication in the *Journal of Physiology*.

Kind regards,

Professor David J. Adams
Distinguished Professor
Molecular Horizons | Faculty of Science, Medicine and Health | Bldg 32
University of Wollongong, Wollongong, NSW 2522 Australia
T +61 2 4239 2264 | E djadams@uow.edu.au

Professor George Mellick
Professor of Neuroscience
Research Leader (Clinical Neuroscience) | Institute for Biomedicine and Glycomics
Griffith University, Nathan, QLD 4111 Australia
T +61 7 3735 5019 | E G.Mellick@griffith.edu.au

Dr. Linlin Ma
Senior Lecturer, Institute for Biomedicine and Glycomics, School of Environmental and
Science, Griffith University, Nathan, QLD 4111 Australia
T +61-7-3735-4175 | E linlin.ma@griffith.edu.au

REVISION

Editor and Reviewer comments:

We sincerely thank the reviewers and editors for their excellent comments and have carefully revised the manuscript. Page numbers below refer to the **Change-tracked manuscript**.

First, the antibodies used to detect Kir4.2 proteins in Western blot and immunoblotting were anti-V5 or anti-FLAG antibodies, instead of anti-Kir4.2 antibodies. This is because the expressed proteins were tagged with V5 and FLAG tags, allowing for highly specific detection of the target protein. We have now highlighted this information in each relevant figure legend. We thank the Senior Editor for bringing up this important point.

Reviewer #1:

1) From the data of double-immunolabeling described in Figure 7, the authors concluded that expression of Kir4.2R28C protein on the plasma membrane was reduced as compared to wild-type Kir4.2 and this change is a major source for loss of K⁺ current in the mutant channel. However, from my standpoint, the reduction of the surface protein is too modest (~10 %) to account for the dramatic suppression of the channel's current. Accordingly, the authors need to add multiple experiments to reinforce their conclusion. Minimum requirement is to isolate the membrane fraction and carry out western blotting with this sample.

Revision: We appreciate this constructive comment. In the first paragraph on page 20, following the description of Fig 7, we state “These results require further verification using alternative methods. Nevertheless, current observations support the hypothesis that the mutation not only affects the stability and turnover of the Kir4.2 channel but also impairs the protein's plasma membrane trafficking capacity once it has been fully processed within the cell.” We are fully aware that immunofluorescence (IF) methodology is just one of the options for detecting membrane-resident proteins. We used the membrane fractionation method to study the membrane trafficking of ion channels (see Ma L. *et al.*, 2009 *Int J Biochem Cell Biol*, DOI: 10.1016/j.biocel.2008.12.006). However, recent advances in confocal imaging and quantitative image analysis techniques have led us to adopt a confocal microscopy-based IF approach. This transition was driven by several key advantages:

First, the combination of confocal microscopy with advanced image analysis provides superior resolution and accuracy. This allows for the detailed visualization of protein localization and distribution within their native cellular context, surpassing the capabilities of traditional methods.

Second, our optimized protocol enables precise quantitative analysis of hundreds of cells. This high-throughput capability significantly improves the statistical robustness of our findings, enabling more reliable conclusions about protein dynamics.

This approach is now widely used for studying cellular protein distribution. For example, the paper recommended by Reviewer #2 (<https://doi.org/10.1074/jbc.M116.729822>) employs a similar methodology.

We agree with you, as well as reviewer #3, that we should further emphasize that the reduced expression, protein stability, and membrane trafficking only partially account for the mutation's loss of function. Therefore, we have modified the discussion on page 26 (highlighted in yellow) as follows:

“Nevertheless, in plasma membrane trafficking and the impaired stability of the mutant channel protein alone do not fully account for the complete loss of channel function. These findings warrant further exploration to elucidate the underlying mechanisms.”

2) In Figure 5, I can understand that MG132, a blocker for UPS-mediated degradation, had little effect on the protein level of Kir4.2WT and Kir4.2R28C. Nevertheless, before making such conclusion, the authors should examine if Kir4.2WT and Kir4.2R28C are ubiquitinated by biochemical approach with a suitable positive control such as KATP channel.

Revision: We thank the Reviewer for this comment. The reviewer is absolutely correct that MG132 inhibits the 26S proteasome, which specifically degrades ubiquitinated proteins. Regarding the proteasome-mediated protein degradation pathways, ion channels are primarily degraded through the ubiquitin-dependent pathway, rather than the ubiquitin-independent pathway (for reference, see the *PNAS* review “The ubiquitin-proteasome pathway: The complexity and myriad functions of proteins death” doi: [10.1073/pnas.95.6.2727](https://doi.org/10.1073/pnas.95.6.2727)). Since our primary objective was to determine whether the ubiquitin-proteasome degradation pathway plays a key role, blocking the downstream step (the proteasome) without observing an effect is sufficient to address our question. To improve accuracy, we refined our descriptions of the results and conclusions to explicitly refer to ubiquitin-proteasome system-mediated degradation. For example, we now state: “Indeed, our findings indicate that Kir4.2 proteins are predominantly degraded via autophagy-lysosomal mechanisms rather than through the ubiquitin-proteasome system” (highlighted in yellow on page 25).

3) In Figure 6B and 6C, I am not convinced by the authors' conclusion described in the text because of large variation of the data for Kir4.2R28C treated with bafilomycinA1 for 6 h. To strengthen the conclusion, I request the authors to increase the data points (n = 6 at minimum).

Revision: We thank the Reviewer for this valuable suggestion. Additional experiments have been conducted, bringing the total to **eight** biological repeats. Please see the updated Fig 6 on page 20. The description of this figure has also been updated on page 19 accordingly (highlighted in yellow), as shown below:

“As shown in Fig 6, prolonged lysosome inhibition led to increased accumulation of Kir4.2^{WT} proteins in the cells, suggesting that this channel protein is primarily degraded through the lysosomal or autophagic pathway ($P = 0.0211$ and 0.0042 for 3 h and 6 h treatments, respectively. $N = 8$, Fig 6B). In contrast, while Kir4.2^{p.R28C} proteins exhibited a similar trend, they were less sensitive to Baf A1 treatment compared to Kir4.2^{WT}, indicating that this mutant may have additional regulatory controls that mitigate the effects of Baf A1 treatment (Fig 6B). Furthermore, Baf A1 had similar effects on both mature and immature Kir4.2 proteins, as the ratio of the upper to lower bands remain unchanged following treatment for both Kir4.2^{WT} and Kir4.2^{p.R28C} (Fig 6C). ”

4. As I mentioned in comment #1, there is a discrepancy between a dramatic loss of the current elicited by Kir4.2R28C channel (Figure 2) and a modest decrease of expression of the protein on the cell surface (Figure 7). This inconsistency suggests that the rest of the channel proteins on the cell surface would be characterized by impairment of single-channel properties. Therefore, the authors need to examine cell-attached patch to examine the change of the parameters such as open time and single-channel conductance.

Revision: We acknowledge that single-channel recordings could provide valuable mechanistic insights. Therefore, we attempted to record single-channel activity in cell-attached patches of HEK293T cells transfected with WT and mutant Kir4.2. However, the substantial endogenous K⁺ channel expression in HEK293T cells (PMIDs: 15206817, 29665277, 35567642) made it difficult to confidently isolate and resolve individual Kir4.2 channel events. This challenge was further compounded by the loss-of-function phenotype of the R28C mutation, which results in infrequent or absent channel openings, limiting the feasibility of obtaining reliable single-channel conductance data.

Chord conductance estimations from whole-cell ramp currents in HEK293T cells expressing Kir4.2-WT and Kir4.2-WT+R28C, measured at membrane potentials where the Ba²⁺-sensitive current was clearly distinguishable (-20 mV to -140 mV), revealed a substantially lower conductance in WT+R28C-expressing cells compared to WT-transfected cells under standard recording conditions (Fig 2C revised manuscript).

These findings indicate that the loss-of-function phenotype of the R28C mutation is associated with reduced macroscopic conductance at the plasma membrane. However, determining whether the severe LOF observed in homomeric Kir4.2-R28C results from protein processing defects alone, impaired single-channel conductance, and/or altered gating will require further investigation.

Minor: In western blot analyses, two fragments are detected in both Kir4.2 WT and Kir4.2R28C. The terminology to indicate these two are confusing (upper/bottom or mature/immature). Throughout the manuscript, the authors should unify this issue.

Revision: We thank the Reviewer for this suggestion. We have now ensured consistent use of the terminology "upper band" and "lower band." When referring to "mature and immature proteins," we explicitly state "mature (upper band) and immature (lower band)" to enhance clarity.

Reviewer #2:

The authors did not explain the dichotomy of how this channel mutant affects neurological functions or the degeneration of dopaminergic neurons if it doesn't express in either neurons or astrocytes. The brain atlas (<https://www.proteinatlas.org/ENSG00000157551-KCNJ15>) and other single-cell studies (<https://brainrnaseq.org/?532754659=2988537778>) do not provide evidence for the meaningful expression of the KCNJ15 channel in neuronal tissue or astrocytes, compared to channels that do express in neurons, like KCNJ6, or astrocytes, like KCNJ10.

In mice, this channel expresses mainly in kidneys proximal tubules to affect bicarbonate reabsorption (<https://doi.org/10.1016/j.kint.2019.09.028>) and glutaminase activity (<https://doi.org/10.1016/j.celrep.2022.111840>).

Revision: We thank the Reviewer for raising this important point. To address this, we searched various databases and literature for the expression profile of Kir4.2. While RNAseq data from broad expression profiling studies provide useful insights, targeted quantification of specific proteins is generally more accurate. A study by Shcherbatyy *et al.* specifically examined the expression pattern of 320 channels in the two-week-old rat brain using nonradioactive robotic *in situ* hybridization and stringently generated coronal brain sections (Shcherbatyy V. *et al.*, 2015, *Neuroinformatics*, DOI: 10.1007/s12021-014-9247-0). This

study characterized Kir channels in detail and reported strong Kir4.2 (*KCNJ15*) expression throughout most brain regions, as summarized in Table 2 of the paper (cited below).

Additionally, we recently conducted an IHC study on WT and *KCNJ15*^{-/-} mouse brain tissues. Our preliminary results indicate that Kir4.2 is expressed in neurons (unpublished data). We are currently working to validate this finding and investigate whether it is also expressed in glial cells.

Table 2 Patterns annotated by gene and structure for *Kcnj* channel family

Structure	Kcnj1	Kcnj2	Kcnj12	Kcnj4	Kcnj14	Kcnj3	Kcnj6	Kcnj9	Kcnj5	Kcnj10	Kcnj15	Kcnj16	Kcnj8	Kcnj11	Kcnj13
	Kir1.1	Kir2.1	Kir2.2	Kir2.3	Kir2.4	Kir3.1	Kir3.2	Kir3.3	Kir3.4	Kir4.1	Kir4.2	Kir5.1	Kir6.1	Kir6.2	Kir7.1
Amygdala	S+	S+	S++	R+++	S++	R+++	R++	S+++	S+	S+	S+++	S++	S+	S++	S+
Basal Forebrain	S+	S+	S++	R+++	S+++	R+++	R++	S+++	S+	S+	S+++	S++	S+	S++	S+
Caudate Putamen	S+	S++	S++	S+++	S++	S++	U+	S+++	S+	S++	S+++	S++	S++	S++	S+
Cerebral Cortex	S+	S+	S++	S+++	S+++	S+++	R+++	S+++	R++	S++	S+++	S+++	S++	S++	S+
Globus Pallidus	S+	S+	S++	S+++	S+++	R+++	S+	S+++	S+	S++	S+++	S++	S++	R+++	S+
Hippocampus	S+	R++	S++	R+++	R+++	R+++	R+++	S+++	S+	S++	S+++	S+++	S++	S++	S+
Hypothalamus	S+	R++	S++	S++	S++	R++	R++	S+++	S+	R++	S+++	S++	S+	S++	S+
Midbrain	S+	S+	S++	R+++	R+++	R+++	R+++	S+++	S+	R++	S+++	S++	S++	S++	S+
Pons	S+	S+	S++	S++	S+++	R++	R+++	S+++	S+	R++	S+++	S+++	S++	S++	S+
Thalamus	S+	R++	R+++	R+++	S+++	R+++	R+++	S+++	S+	R++	S+++	S+++	S++	S++	S+
Septum	S+	S+	S++	S++	S++	R++	R++	S+++	S+	S+	S+++	S++	S+	S++	S+
Ventricles	S+	S+	S+	S+	S+	U+	U+	S++	S+	R+	R++	R++	S+	S+	R+++
Fiber Tracts	S+	S+	S+	S+	S+	S+	S+	S++	S+	S+	S+	S+	S++	S+	S+

R regional, S scattered, U ubiquitous; +++ strong (red); ++: moderate (blue); +: weak (yellow)

2) The authors should discuss the role of this residue in channel folding. There are ample structures of IR channels that can provide a good insight into the function of this fully conserved residue. Furthermore, R28 was identified as part of an AP-1 clathrin adaptor-dependent Golgi export signal (<https://doi.org/10.1074/jbc.M116.729822>); the authors should discuss their findings considering these findings.

Revision: We thank the Reviewer for drawing our attention to this interesting study by Li X *et al.* Although this paper does not specifically address Kir4.2, it is insightful to learn that the equivalent Arg in Kir4.1 (R29), a close homolog of Kir4.2, plays an important role in Golgi export. In light of this, we have revised the discussion on how the R28C mutation may affect Kir4.2's membrane trafficking, referencing this study on page 26 (highlighted in yellow):

“Notably, the residue equivalent to R28 in Kir4.2's homolog, Kir4.1 (R29), has been shown to contribute to an AP-1 clathrin adaptor-dependent Golgi export signal (27RRR29). Mutation of these three arginine residues on the N-terminus of Kir4.1 to alanine results in Golgi retention of the protein and significantly reduced membrane expression (Li et al. 2016). It would be interesting to investigate whether Kir4.2 shares a similar Golgi export signal patch as Kir4.1.”

Minor: It is very difficult to appreciate the expression pattern/level localized to the plasma membrane. Can the authors provide an expanded view of one cell in addition to the images in Fig. 7?

Revision: We thank the reviewer for pointing this out. An expanded view of several representative cells has been added to Fig 7A for both Kir4.2^{WT} and Kir4.2^{R28C}.

Reviewer #3:

1) Page 14 Line 1; “There was not statistical difference between the PNGase F-treated Kir4.2WT and Kir4.2R28C proteins, while with Endo H treatment, Kir4.2R28C exhibited a level approximately 50% of that observed for Kir4.2WT (Fig 3E)”

The experiment was done with harvested cells that were cultured under the same condition. Hence, if the protease inhibitor treatment was tight during the experiment, theoretically the total protein level of WT (and of R28C) among the three tested conditions should be identical. If they are different, it is due to protein degradation happened in the middle of the experiments or the analyses were not accurate. The difference mentioned in the above sentence arose due to the fact that the amount of WT protein in the PNGase F treated sample got decreased than the two other conditions (Non-treated and Endo H treated). Hence, the sentence does not support any mechanisms and is likely incorrect. So, it is recommended to exclude the sentence from the manuscript.

Revision: We thank the Reviewer for this important point. Fig 3E was included to assess whether removal of glycans by PNGase and EndoH affects the protein’s stability and solubility, potentially making it more susceptible to proteolytic enzymes present during the processing. We agree that the current description of this figure may be unclear and have revised the explanation on pages 12-13 (highlighted in yellow) as follows:

“Neither PNGase F nor Endo H altered the overall protein levels of Kir4.2^{WT} or Kir4.2^{R28C} (Fig. 3E). This indicates that removal of glycans from the protein does not impact the stability or solubility of Kir4.2, nor does it make the protein susceptible to proteolysis during the experimental process.”

2) Page 14. Last paragraph: “Interestingly, Kir4.2WT and Kir4.2 R28C exhibited differing responses to tunicamycin treatment. Kir4.2WT’s overall expression and the ratio between mature and immature proteins were not significantly altered by tunicamycin (Fig 4B&4C).”

The argument is not supported by the actual experimental data shown in Fig 4A. Fig 4A shows that 6h treatment tunicamycin caused the total expression level of both the WT and the mutant increased and the relative ratio between the top and the bottom band flipped, indicating newly synthesized proteins could not be glycosylated, which was the same for both proteins. (Also please see the minor revision point below regarding this figure.) Hence, the actual data is in contrast to their argument. It seems like that one out of 3 replicate experiments showed no increase in the protein level and no increase in the un-glycosylated (immature) protein for the WT protein after 6hrs tunicamycin treatment. The one data point actually abolished the statistical significance. In my humble opinion, overall, the data indicate that both proteins responded to the tunicamycin treatment in the same manner. A few more replicates are needed to consolidate their argument, or the data figure should be replaced with the one matching the argument.

Revision: We appreciate the reviewer’s advice. We have now conducted additional experiments, increasing the number of biological repeats to **eight**. Please see the updated Fig 6 on page 20. The description of this figure has been revised accordingly on page 19 (highlighted in yellow), as follows:

“As shown in Fig 6, prolonged lysosome inhibition led to increased accumulation of Kir4.2^{WT} proteins in the cells, suggesting that this channel protein is primarily degraded through the lysosomal or autophagic pathway ($P = 0.0211$ and 0.0042 for 3 h and 6 h treatments, respectively; $N = 8$, Fig 6B). In contrast, while Kir4.2^{p.R28C} proteins exhibited a similar trend, they were less sensitive to Baf A1 treatment compared to Kir4.2^{WT}, indicating that this mutant may have additional regulatory controls that mitigate the effects of Baf A1 treatment (Fig 6B). Furthermore, Baf A1 had similar effects on mature and immature Kir4.2 proteins, as the ratio of the upper to lower bands remained unchanged with the treatment for both Kir4.2^{WT} and Kir4.2^{p.R28C} (Fig 6C). ”

3) Page 17. Line 8 from the bottom: “Although not as pronounced as the increase in LC3B-II, a discernible trend showed that the overall levels of both Kir4.2WT and Kir4.2R28C rose with extended Baf A1 exposure (Fig. 6B). Notably, the baseline level of Kir4.2R28C was approximately 50% that of Kir4.2WT, yet this difference vanished following a 6-hour Baf A1 Treatment (Fig. 6B).” This argument is not supported by the actual data shown in Fig 6A. essentially there is no changes in protein levels after Baf A1 treatment for both WT and R28C mutant proteins. More repeats are needed to support their argument, or the data figure should be replaced with the one matching the argument.

Revision: We thank the reviewer for this advice. This issue has now been addressed by including a new representative figure in Fig 6A and increasing the number of biological repeats (as noted above).

4) Figure 7. They used colocalization imaging to assess membrane expression of the proteins. But it may be more reliable and quantitative to use the biotinylation and membrane fractionation method.

Revision: We thank the reviewer for this suggestion. Please refer to our response to the first question from Reviewer #1.

5) Page 23, First paragraph: “Given that the expression of both Kir4.2WT and Kir4.2R28C was driven by the same promoter under identical conditions using model cell lines, it is more likely that this differential expression is caused by post-translational modifications, protein folding and turnover issues, which are crucial for the protein stability and function.”.

This presumption is not automatically true since the protein level can be lower due to slowed translation and also reduced mRNA stability due to mutations. So, there are other mechanisms how protein level is kept lower than the other.

“Interestingly, similar to the dF508 mutation on the N-terminal tail of CFTR, the mutant channel Kir4.2R28C also exhibits decreased stability and longevity in cells compared to Kir4.2WT (Fig 4).” I cannot figure how Fig 4 supports this argument. Could you elaborate?

Revision: We appreciate this valuable advice from the reviewer. We agree that while unlikely, we cannot entirely rule out the potential impact of this mutation on transcription and translation. To address this, we have revised the relevant section (highlighted in yellow on page 24) as follows:

“The expression of both Kir4.2^{WT} and Kir4.2^{R28C} was driven by the same promoter under identical conditions using model cell lines. Although a single mutation can potentially influence transcription and translation efficiency under this condition, it is more likely that the observed differences in expression result from post-translational modifications, protein folding, and turnover dynamics, which are critical for protein stability and function.”

Additionally, the statement “Kir4.2^{R28C} also exhibits decreased stability and longevity in cells compared to Kir4.2^{WT} (Fig 4).” is based on the results presented in the section titled “The stability and longevity of the Kir4.2^{R28C} mutant are reduced relative to Kir4.2^{WT}” (page 15-16), which primarily discusses the findings in Fig 4.

Briefly, tunicamycin primarily disrupts new glycosylation events, potentially disrupting proper protein folding and function, leading to cellular stress, activation of the unfolded protein response (UPR), and engagement of stress pathways. Notably, the mutant proteins, but not the WT proteins, exhibited a significant increase in overall protein levels and a notable decrease in the proportion of mature proteins following 6 h of tunicamycin treatment. This suggests that WT proteins may be more resilient to disruptions

in glycosylation, potentially due to their greater inherent stability and/or longer half-life compared to the mutant variants.

6) Page 23. Second paragraph: “Indeed, we found that Kir4.2 proteins are not primarily subjected to the ubiquitine-proteasome system mediated degradation(Fig5), but are more susceptible to the autophagy-lysosomal degradation pathway (Fig 6).”

The first half is agreeable, but the second half is not. I cannot see any differences between the two proteins in Fig 6 in their response to the Baf A1 treatment.

Revision: We appreciate the reviewer’s comment. As mentioned above, we have now conducted eight biological repeats and have updated Fig. 6A to better represent the results.

Minor:

1) Figure 4. It is unclear if the total expression time among the three conditions were identical for this experiment. Was the drug treatment started 6hrs, 2 hrs, and 0 hrs before the harvest so that the total length of time of protein expression is identical?

Since the experiment was done with transient transfection, the total amount of protein expression should increase as the incubation time elongates unless the treatment was done after the protein expression reached the steady state after the transfection. So I just want to double check if the increased total protein level of WT and R28C after 6 hour treatment was simply due to the cells were cultured for a longer time.

The tick label in Fig 4B and 4C should be fixed to be 2h instead of 3h.

Page 6: method for western blot: it should be 24-well plate instead of 25-well plate

Revision: We appreciate the reviewer’s comments. Yes, drug treatments were consistently applied throughout the study to ensure uniform protein expression duration across all conditions. For example, in the 6 h treatment condition, the cells were treated for six hours before harvesting, which was performed simultaneously with the 0 h treatment condition.

We apologize for the typographical errors. They have now been corrected.

2) Page 24: the last part of the first paragraph: “Particularly, the Kir4.2R28C mutant disrupts a PKB phosphorylation recognition site, a modification previously reported to affect the assembly and stability of the potassium channel/scaffold protein complex through phosphorylation.”

Tanemoto et al 2002 reports PKA phosphorylation of the serine residue at the very C-terminus of Kir5.1 affect its interaction with PSD95 proteins. Hence, it is hard to conceive how R28C would disrupt PKB phosphorylation recognition site. Please remove this sentence or elaborate in detail how this argument arose.

Revision: We thank the reviewer for this comment. Our analysis using the NetPhos database (<https://services.healthtech.dtu.dk/service.php?NetPhos-3.1>) predicted that the R28 residue is located within an Akt/PKB minimum consensus sequence (RXRXXS/T) in Kir4.2, as illustrated in the following figure. However, as this prediction is based solely on a minimum consensus sequence without experimental verification, we agree to remove the sentence.

Kir4.2				Kir4.2 ^{R28C}			
Sequence context	Score	Kinase	Prediction	Sequence context	Score	Kinase	Prediction
PRVMSKSGH	0.994	unsp	YES	PCVMSKSGH	0.808	unsp	YES
PRVMSKSGH	0.774	PKB	YES	PCVMSKSGH	0.457	GSK3	.
PRVMSKSGH	0.518	RSK	YES	PCVMSKSGH	0.456	PKC	.
PRVMSKSGH	0.467	CaM-II	.	PCVMSKSGH	0.451	CaM-II	.
PRVMSKSGH	0.462	GSK3	.	PCVMSKSGH	0.431	cdc2	.
PRVMSKSGH	0.457	PKA	.	PCVMSKSGH	0.398	CKI	.
PRVMSKSGH	0.457	PKG	.	PCVMSKSGH	0.380	DNAPK	.
PRVMSKSGH	0.455	PKC	.	PCVMSKSGH	0.341	RSK	.
PRVMSKSGH	0.426	cdc2	.	PCVMSKSGH	0.331	PKG	.
PRVMSKSGH	0.387	CKI	.	PCVMSKSGH	0.329	CKII	.
PRVMSKSGH	0.380	DNAPK	.	PCVMSKSGH	0.304	p38MAPK	.
PRVMSKSGH	0.300	cdk5	.	PCVMSKSGH	0.283	ATM	.
PRVMSKSGH	0.282	p38MAPK	.	PCVMSKSGH	0.273	PKB	.
PRVMSKSGH	0.280	ATM	.	PCVMSKSGH	0.265	cdk5	.
PRVMSKSGH	0.260	CKII	.	PCVMSKSGH	0.213	PKA	.

3) Page 24: the last paragraph: “Our findings reveal that this mutation leads to abolished channel function, diminished overall expression, and decreased stability of the Kir4.2 protein. Furthermore, there is an increased likelihood of lysosomal degradation and compromised plasma membrane trafficking of the mutant channel.”

“This mutation leads to abolished channel function, diminished overall expression” this part is fully supported by the experimental results in the manuscript. However, the remaining part is not convincing. In order to get support for these arguments, more repeats are required for Fig 4 and 6. And Fig 7 can be further supported by the biotinylation and membrane fractionation experiment.

Revision: We thank the reviewer for this comment. As mentioned above, additional biological repeats have been conducted for Fig. 4 (six repeats) and Fig. 6 (eight repeats). Consequently, we have updated this part of the summary (on page 25) as follows:

“Our findings show that this mutation abolishes channel function with significant dominant-negative effects, reduced overall expression, and decreased stability of the Kir4.2 protein. Furthermore, the mutant channel exhibits impaired plasma membrane trafficking capacity.”

Thank you kindly for your consideration.

Best wishes,

David J. Adams, George Mellick & Linlin Ma
 University of Wollongong, Wollongong & Griffith University, Brisbane, AUSTRALIA.

Dear Dr Ma,

Re: JP-RP-2025-287046R1 "Parkinson's Disease-Linked Kir4.2 Mutation R28C Leads to Loss of Ion Channel Function" by Xiaoyi Chen, Rocio K. Finol-Urdaneta, Mo Chen, Alex Sykes, Bingmiao Gao, Jamila Iqbal, David J. Adams, George D Mellick, and Linlin Ma

Thank you for submitting your manuscript to The Journal of Physiology. It has been assessed by a Reviewing Editor and by 2 expert referees and we are pleased to tell you that it is potentially acceptable for publication following satisfactory major revision.

LANGUAGE EDITING AND SUPPORT FOR PUBLICATION: If you would like help with English language editing, or other article preparation support, Wiley Editing Services offers expert help, including English Language Editing, as well as translation, manuscript formatting, and figure formatting at www.wileyauthors.com/eoo/preparation. You can also find resources for Preparing Your Article for general guidance about writing and preparing your manuscript at www.wileyauthors.com/eoo/prepresources.

REVISION CHECKLIST:

We look forward to receiving your revised submission.

Yours sincerely,

Peying Fong
Senior Editor
The Journal of Physiology

REQUIRED ITEMS

- Please include an Abstract Figure file, as well as the Figure Legend text within the main article file. The Abstract Figure is a piece of artwork designed to give readers an immediate understanding of the research and should summarise the main conclusions. If possible, the image should be easily 'readable' from left to right or top to bottom. It should show the physiological relevance of the manuscript so readers can assess the importance and content of its findings. Abstract Figures should not merely recapitulate other figures in the manuscript. Please try to keep the diagram as simple as possible and without superfluous information that may distract from the main conclusion(s). Abstract Figures must be provided by authors no later than the revised manuscript stage and should be uploaded as a separate file during online submission labelled as File Type 'Abstract Figure'. Please also ensure that you include the figure legend in the main article file. All Abstract Figures should be created using BioRender. Authors should use The Journal's premium BioRender account to export high-resolution images. Details on how to use and access the premium account are included as part of this email.

EDITOR COMMENTS

Reviewing Editor:

In this revised manuscript, the authors performed additional experiments to increase the number of replicates in Fig. 4 and Fig. 6. However, referee #3 feels that this revision doesn't provide supported results to fully explain the complete loss of channel function of R28C mutant. The authors should address this key point (see referee #3's comments) by additional experiments or revision of conclusions.

Please also see 'Required Items' above.

Senior Editor:

Review of your revised manuscript, "Parkinson's Disease-Linked Kir4.2 Mutation R28C Leads to Loss of Ion Channel Function" is now complete. Two of the three original Expert Referees (1 and 3) provide their assessments of this latest version. You will also find a summarized report from the Reviewing Editor. In general, they remark on the study's potential for generating interest in The Journal of Physiology's readership. They appreciate the overall improvement in robustness of the conclusions, primarily resulting from increased reproducibility.

Nonetheless, the second attached Referee report (from the original Referee 3) raises an important point that is echoed by the Reviewing Editor. This point pertains to the implication (by omission) that the mutations studied do not exert its effects by abolishing channel activity, as the electrophysiological demonstration of activity is limited to whole cell current measurements. The Referee notes that the reported studies do not rule out such a contribution and provides reasons why it may, in fact, be likely. For completeness, such an alternative explanation formally should be ruled out. If not demonstrable by single channel measurements (which would entail additional experiments), please consider a more measured phrasing of

precisely what the present findings do (do not) signify, ideally within the Discussion.

Perhaps it is even possible that all three levels of disrupted function: single channel properties, stability, and trafficking, may come into play?

Finally, I am unable to find information about a critical reagent used in the western blot analysis, i.e. the anti-Kir 4.2 (KCNJ15) antibody used in performing these blots. Information for such an antibody does show up in the Methods section pertaining to immunofluorescence studies. If it is the same antibody, it should be stated in the preceding section on the immunoblotting/western blotting. This is important to know, since not all antibodies are useful for the detection across assay types. If it is indeed the same, then there should be sufficient confirmatory data offered for review purposes.

REFeree COMMENTS

Referee #1:

In this revision the authors have sufficiently addressed the comments and concerns that I raised. I have no more comments.

Referee #3:

In this manuscript, it was clearly demonstrated that the Kir4.2 R28C mutant showed a dominant negative effect (Fig 2). The authors investigated whether cellular proteostasis is responsible for the loss of function phenotype. The key observations include:

The total protein level of the R28C mutant was approximately 50% of the WT (Fig 3A, B).

Glycosylation patterns were indistinguishable between the two proteins (Fig 3C-F).

Inhibiting proteasome activity did not increase the protein levels (Fig 5).

Inhibiting lysosomal degradation mechanisms elevated protein levels (WT > R28C).

The relative membrane fractions expressing Kir4.2 were ~30% for the WT and ~25% for the mutant out of the membrane fraction expressing Na/K-ATPase proteins.

Based on these observations, it was concluded that the R28C mutant destabilizes the protein and decreases surface expression, leading to the loss of function phenotype.

However, the differences between the WT and R28C mutant proteins in terms of surface expression levels and their responses to different treatments (tunicamycin, MG132, and Baf A1) are relatively small. These small differences cannot fully explain the complete loss of channel activity and the dominant negative effect observed in the mutant proteins. Reviewer 1 rightly pointed this out in the first round of review. However, without proper experimental support (chord conductances (Fig. 2C) are essentially representing the same data shown in the macroscopic currents during the ramp protocol (Fig. 2A)), the authors excluded the possibility that the mutant can abolish the channel activity, which is actually highly likely.

R28 is 100% conserved among all Kir channels, indicating the critical importance of arginine at this position. The deep mutational scanning assay conducted on human Kir2.1 (<https://elifesciences.org/articles/76903>) provides insightful data regarding this conserved arginine. Mutating the arginine to any other residue except histidine caused a significant reduction in surface expression. Surprisingly, the arginine-to-cysteine mutation was the most tolerated next to the arginine-to-histidine mutation in terms of surface expression. Despite this, the arginine-to-cysteine mutation resulted in a significant reduction in channel activity, indicating that this mutation likely affects channel activity in addition to trafficking.

Even though the residues are not close to the pore, the arginine forms a salt bridge with the neighboring glutamate, connecting the N-terminal peptide (that is structured; this Arg is not unstructured, which was argued by the authors) to the C-terminal bottom of the cytoplasmic domain. Inter-subunit salt bridges at the bottom of the CTD are important for stabilizing the active states of Kir channels (<https://doi.org/10.1085/jgp.201611719>), making it unlikely that the mutation won't affect

channel activity.

Therefore, it is highly recommended to perform patch-clamp experiments to confirm the channel properties. If this is not possible, please tone down the results and discussion to clarify that the changes in proteostasis are marginal and revise the manuscript to not suggest that the R28C mutation won't affect individual channel activity.

Minor:

Fig 1B: Gly -> Glu, also label on other residues interacting with R28 will be helpful.

It was claimed that Pro27 is hydrogen bonded to R28 in the manuscript, which is somewhat doubtful.

END OF COMMENTS

Professor Peying Fong
Senior Editor
Journal of Physiology

Re: Manuscript Number: JP-RP-2024-287046 "Parkinson's Disease-Linked Kir4.2 Mutation R28C Leads to Loss of Ion Channel Function"

Dear Professor Fong,

Thank you for your email regarding our revised manuscript submitted to the *Journal of Physiology*, and for the opportunity to resubmit following major revisions.

We sincerely appreciate the reviewer's thoughtful and constructive feedback, which has been invaluable in strengthening our manuscript. We are especially grateful for the time and expertise invested in the review process.

In response to Reviewer #3's comments, we have further revised the manuscript and provide a detailed point-by-point response below. We hope that the updated version will now meet the journal's standards for publication.

Kind regards,

Professor David J. Adams
Distinguished Professor
Molecular Horizons | Faculty of Science, Medicine and Health | Bldg 32
University of Wollongong, Wollongong, NSW 2522 Australia
T +61 2 4239 2264 | E djadams@uow.edu.au

Professor George Mellick
Professor of Neuroscience
Research Leader (Clinical Neuroscience) | Institute for Biomedicine and Glycomics
Griffith University, Nathan, QLD 4111 Australia
T +61 7 3735 5019 | E G.Mellick@griffith.edu.au

Dr. Linlin Ma
Senior Lecturer, Institute for Biomedicine and Glycomics, School of Environmental and
Science, Griffith University, Nathan, QLD 4111 Australia
T +61-7-3735-4175 | E linlin.ma@griffith.edu.au

REVISION

Response to Editor and Reviewer comments

We sincerely thank the reviewers and editors for their thoughtful and constructive feedback, which has significantly improved our manuscript. We have carefully revised the manuscript in response to each comment. Page numbers referenced below refer to the **Change-tracked version of the manuscript**.

General Clarification – Antibody Usage

We appreciate the Senior Editor’s comment regarding antibody specificity. To clarify, all Kir4.2 constructs used in this study included both 3xFLAG and a V5 tags, enabling the use of anti-FLAG and anti-V5 antibodies for specific and reliable detection in both Western blotting and immunofluorescence experiments. Accordingly, anti-Kir4.2 antibodies were not used. This detail has now been explicitly stated in the relevant figure legends.

Reviewer #1

We are grateful that Reviewer #1 is satisfied with our previous revisions.

Reviewer #3:

It is highly recommended to perform patch-clamp experiments to confirm the channel properties. If this is not possible, please tone down the results and discussion to clarify that the changes in proteostasis are marginal and revise the manuscript to not suggest that the R28C mutation won't affect individual channel activity.

We thank the reviewer for this important and insightful recommendation.

As noted in the previous version of the manuscript (page 25), we acknowledged that the modest impairments in proteostasis and membrane trafficking do not fully account for the complete loss of channel function observed with the R28C mutation:

“Nevertheless, the modest reduction in plasma membrane trafficking and the impaired stability of the mutant channel protein alone do not fully account for the complete loss of channel function.”

Furthermore, the possibility that N-terminal mutations can profoundly affect channel function was also discussed (page 23):

“While a mutation in the N-terminal cytoplasmic tail may seem counterintuitive to have such a profound impact, disruptions in channel gating, trafficking, or regulatory mechanisms caused by such mutations are common across various ion channels.”

As single-channel patch-clamp recordings are not feasible with our current setup for reasons as previously outlined, we have further revised the manuscript to emphasize the limitations of our findings and integrate Reviewer #3’s feedback. Please refer to the updated discussion on pages 25–26:

“Nevertheless, the modest reduction in plasma membrane trafficking and the impaired stability of the mutant channel protein alone do not fully account for the complete loss of channel function. Although it may seem counterintuitive, mutations in the N-terminal cytoplasmic tail can profoundly affect channel function, as disruptions in gating and regulatory mechanisms are well documented across ion channel families. For instance, the $\Delta F508$ mutation (deletion of Phe508) in the cystic fibrosis transmembrane

conductance regulator (CFTR) protein disrupts folding and maturation, resulting in near-complete loss of function (Lukacs *et al.* 1993; Lukacs *et al.* 1994; Jensen *et al.* 1995). Similarly, the R104W mutation in the N-terminus of Nav1.5 abolishes Na⁺ currents and exerts a dominant-negative effect, contributing to Brugada syndrome (Clatot *et al.* 2012; Doisne *et al.* 2021).

Notably, R28 is fully conserved across all Kir channels, underscoring its critical functional importance. A deep mutational scanning study of human Kir2.1 revealed that the equivalent R46C mutation markedly disrupts channel activity (Coyote-Maestas *et al.* 2022), consistent with structural evidence showing that R46 in Kir2.1 forms a stabilizing salt bridge between the N-terminal and the C-terminal cytoplasmic domains. Disrupting such inter-subunit interaction is known to impair channel function (Borschel *et al.* 2017). Further studies are required to elucidate the precise mechanisms of R28C-mediated dysfunction in Kir4.2. ”

Minor: Fig 1B: Gly -> Glu, also label on other residues interacting with R28 will be helpful. It was claimed that Pro27 is hydrogen bonded to R28 in the manuscript, which is somewhat doubtful.

Revision: We thank the reviewer for pointing out this oversight and apologize for the error.

The label for Gly has been corrected to Glu in Figure 1B. Additionally, the description of hydrogen bonding has been updated following re-examination of the structural data. The revised text now states: “Arg28 is poised to form hydrogen bonds with Asn37 and Glu304.”

Once again, we thank the reviewers and editors for your constructive feedback and kind consideration.

Best wishes,

David J. Adams, George Mellick & Linlin Ma
University of Wollongong, Wollongong & Griffith University, Brisbane, Australia.

Dear Dr Ma,

Re: JP-RP-2025-287046R2 "Parkinson's Disease-Linked Kir4.2 Mutation R28C Leads to Loss of Ion Channel Function" by Xiaoyi Chen, Rocio K. Finol-Urdaneta, Mo Chen, Alex Sykes, Bingmiao Gao, Jamila Iqbal, David J. Adams, George D Mellick, and Linlin Ma

We are pleased to tell you that your paper has been accepted for publication in The Journal of Physiology.

Yours sincerely,

Peying Fong
Senior Editor
The Journal of Physiology

If you would like to receive our 'Research Roundup', a monthly newsletter highlighting the cutting-edge research published in The Physiological Society's family of journals (The Journal of Physiology, Experimental Physiology, Physiological Reports, The Journal of Nutritional Physiology and The Journal of Precision Medicine: Health and Disease), please click this link, fill in your name and email address and select 'Research Roundup':
<https://www.physoc.org/journals-and-media/membernews>

- You can help your research get the attention it deserves! Check out Wiley's free Promotion Guide for best-practice recommendations for promoting your work at: www.wileyauthors.com/eeo/guide. You can learn more about Wiley Editing Services which offers professional video, design, and writing services to create shareable video abstracts, infographics, conference posters, lay summaries, and research news stories for your research at: www.wileyauthors.com/eeo/promotion.

EDITOR COMMENTS

Reviewing Editor:

The authors have satisfactorily addressed Referee#3's concerns in this revised manuscript.

Senior Editor:

Thank you for responding to points raised in the last round of review. At this time, I am pleased to extend congratulations. Thank you for favoring The Journal of Physiology with this potentially impactful study.

REFEREE COMMENTS

Referee #3:

The manuscript has been revised to address the reviewers' and editors' concerns.